# Neural crest cell-derived DKK1 and NEDD4 modulate Wnt signalling in the second heart field to orchestrate outflow tract development

Sophie Wiszniak [1,2] ✉, Dimuthu Alankarage[3,4], Iman Lohraseb[1], Ceilidh Marchant[1,2], Genevieve Secker[1,2], Deepti Domingo[1,2], Jasmine Hartmann[1,2], Tianyang Zhang [1], Wendy Parker[1,5], John Toubia [1,5], Melissa White[6,7], Sandra Piltz [6,7], Markus Tondl[1], Eleni Giannoulatou [3,8], David Winlaw [9,10], Gillian M. Blue [11,12], Congenital Heart Disease Synergy Group*, Patrick P. L. Tam [13,14], Paul Thomas [6,7], Natasha Harvey [1,2], Sally L. Dunwoodie [3,8] & Quenten Schwarz [1,2] ✉

Cardiac outflow tract morphogenesis requires coordinated interactions between multiple cell populations and is dependent on the contribution of cardiac progenitors from the second heart field. While neural crest cells have been proposed to impact second heart field development, how they regulate progenitor behaviour remains unclear. Here, we discover neural crest cells are a primary source of Dickkopf-1 (DKK1) which modulates Wnt signalling activity in the second heart field to influence the balance between cardiac progenitor maintenance and differentiation. We show that the ubiquitin ligase NEDD4 regulates DKK1, with disruption of *Nedd4* leading to outflow tract defects. We further identify a new NEDD4 variant underlying human congenital heart disease. Our findings uncover an unexpected role for neural crest cells as a rheostat of Wnt signalling in cardiac progenitors, identifying a new molecular pathway promoting outflow tract morphogenesis, and a new causative factor of congenital heart disease.

The cardiac outflow tract of the heart provides the pathway for blood to leave the ventricles and circulate to the body via the aorta, or to the lungs via the pulmonary artery. Outflow tract development is complex and requires the contribution of cell types from multiple embryological origins, including second heart field cardiac progenitor cells, cardiomyocytes, smooth muscle, endocardium, and neural crest cells[1]. During early development, the primitive heart tube expands through the progressive addition of cardiac progenitors from the

[1]Centre for Cancer Biology, University of South Australia and SA Pathology, Adelaide, SA, Australia. [2]College of Health, Adelaide University, Adelaide, SA, Australia. [3]Victor Chang Cardiac Research Institute, Darlinghurst, NSW, Australia. [4]University of New South Wales, Kensington, NSW, Australia. [5]ACRF Cancer Genomics Facility, SA Pathology, Adelaide, SA, Australia. [6]School of Biomedicine and Robinson Research Institute, Faculty of Health and Medical Sciences, University of Adelaide, Adelaide, SA, Australia. [7]South Australian Genome Editing Facility, South Australian Health & Medical Research Institute, Adelaide, SA, Australia. [8]School of Clinical Medicine, Faculty of Medicine and Health, UNSW, Sydney, NSW, Australia. [9]Heart Center, Ann and Robert H. Lurie Children's Hospital of Chicago, Chicago, IL, USA. [10]Feinberg School of Medicine, Northwestern University, Chicago, IL, USA. [11]Heart Centre for Children, The Children's Hospital at Westmead, Westmead, NSW, Australia. [12]Sydney Medical School, Faculty of Medicine and Health, University of Sydney, Sydney, NSW, Australia. [13]Children's Medical Research Institute, University of Sydney, Westmead, NSW, Australia. [14]School of Medical Sciences, Faculty of Medicine and Health, University of Sydney, Sydney, NSW, Australia. *A list of authors and their affiliations appears at the end of the paper. ✉e-mail: Sophie.wiszniak@adelaide.edu.au; Quenten.schwarz@adelaide.edu.au

anterior second heart field, which give rise to the myocardium of the outflow tract and right ventricle. As the outflow tract lengthens, it repositions to overlie the future interventricular septum. Inadequate addition of second heart field cells to the outflow tract leads to insufficient lengthening and underpins conotruncal defects[2].

In the developing heart, neural crest cells form components of the outflow tract valves, arteries, and conotruncal septum. Classical defects associated with disrupted cardiac neural crest cell development include persistent truncus arteriosus, where the outflow tract fails to septate into the aorta and pulmonary arteries[3]. Prior to entering the outflow tract, neural crest cells migrate in close apposition to cardiac progenitors of the second heart field, and are suggested to play a role in modulating the growth dynamics of second heart field derivatives[2]. However, prior models in which neural crest cells have been surgically or genetically ablated[4–7] have not been amenable to elucidate how neural crest cells normally interact with the second heart field to control its growth, differentiation, and morphogenesis.

NEDD4 is an E3 ubiquitin ligase that targets specific protein substrates for ubiquitination to modulate protein function and/or turnover[8]. Prior studies have determined a necessary role for *Nedd4* in heart development, with full knockout mouse models exhibiting DORV and endocardial cushion defects[9]. However, the cell-type-specific mechanisms by which *Nedd4* regulates cardiac development remained unexplored.

In this work, we determine an essential tissue-specific role for *Nedd4* in neural crest cells for correct outflow tract development. *Wnt1-Cre; Nedd4^{fl/fl}* embryos exhibit outflow tract defects typically associated with failed addition of second heart field cells to the outflow tract, suggesting a novel role for *Nedd4* and cardiac neural crest cells in regulating second heart field development. Furthermore, we identify DKK1 as a previously unknown substrate of NEDD4-mediated ubiquitination, and find that this functionality is impaired in a human variant of NEDD4 associated with the outflow tract defect Tetralogy of Fallot. We propose NEDD4 controls DKK1 protein levels in neural crest cells, which in turn modulates Wnt signalling in the second heart field to balance cardiac progenitor maintenance versus myocardial differentiation. This provides new insight into the mechanisms underpinning normal outflow tract morphogenesis, and defines a potential pathogenic mechanism underlying congenital heart disease.

## Results

### *Nedd4* is required in neural crest cells for outflow tract development

Prior studies have reported heart defects in *Nedd4^{−/−}* embryos[9]. We specifically assessed outflow tract defects in *Nedd4^{−/−}* embryos, revealing defects at full penetrance with variable phenotypes, most commonly persistent truncus arteriosus (PTA) and transposition of the great arteries (TGA) (Fig. 1A, E, I–L, Q). *Nedd4* is broadly expressed throughout all embryonic tissues that contribute to early heart development (Supplementary Fig. 1). To determine the cell types in which *Nedd4* activity is critically required, we conditionally removed *Nedd4* using tissue-specific Cre-drivers for cell lineages important in outflow tract development. Ablation of *Nedd4* in the second heart field (*Mef2cAHF-Cre; Nedd4^{fl/fl}*) and endothelial/endocardial cells (*Tie2-Cre; Nedd4^{fl/fl}*) had no discernible impact on the development of the outflow tract (Fig. 1C, D, G, H). In contrast, ablation of *Nedd4* in the neural crest cells (*Wnt1-Cre; Nedd4^{fl/fl}*) led to fully penetrant outflow tract defects such as double-outlet right ventricle (DORV) and overriding aorta (Fig. 1B, F, M–P, Q). Importantly, the outflow tract defects identified in *Nedd4^{−/−}* and *Wnt1-Cre; Nedd4^{fl/fl}* embryos are of the same class of outflow tract defect, that being artery-ventricle alignment defects, that present on a variable spectrum of severity. These findings point to a critical cell-type-specific role for *Nedd4* in neural crest cells to orchestrate outflow tract development, consistent with previous demonstration of essential roles for *Nedd4* in neural crest cells in

craniofacial morphogenesis and peripheral nervous system development[10–12].

### Loss of *Nedd4* in neural crest cells induces premature second heart field differentiation

The myocardial component of the outflow tract is derived from cardiac progenitors of the second heart field, with growth of the outflow tract dependent on the continued addition of cardiac progenitors to lengthen the outflow tract, allowing for heart looping and outflow tract rotation to establish correct artery-ventricle alignment[1,2]. Underpinning the artery-ventricle alignment defects in *Wnt1-Cre; Nedd4^{fl/fl}* embryos, defective outflow tract lengthening at E10.5 (Fig. 2A and Supplementary Fig. 2) led to incomplete clockwise rotation of the outflow tract at E12.5, and mispositioning of the aorta over the right ventricle (Fig. 2B). Semilunar valve morphology was also altered (Fig. 2B). Such artery-ventricle alignment defects typically stem from failed addition of second heart field cells to the lengthening outflow tract. Since the removal of *Nedd4* in second heart field derivatives (*Mef2cAHF-Cre; Nedd4^{fl/fl}*) did not lead to such outflow tract defects (Fig. 1C, G), this suggests a novel role for *Nedd4* and cardiac neural crest cells in regulating second heart field development.

Lineage tracing of neural crest cells using an EGFP reporter revealed that prior to entering the outflow tract, cardiac neural crest cells migrate in close proximity to the second heart field in E9.5 embryos (Fig. 3A, B). This spatial relationship presents an appropriate environment for neural crest cells to interact with the second heart field during development. It is noted that *Wnt1-Cre; Nedd4^{fl/fl}* embryos did not exhibit any deficiency in cardiac neural crest cells (Fig. 3C), nor were changes in cell proliferation (Fig. 3D) or rate of cell death (Fig. 3E) observed in the tissue domain populated by the neural crest cells and second heart field tissue. Furthermore, expression of the second heart field marker Isl1 was unaffected (Fig. 3F and Supplementary Fig. 3C), as were additional markers of the second heart field and other cardiac components (Supplementary Fig. 3), suggesting the development of other cardiogenic tissues was not affected by loss of *Nedd4* function in neural crest cells. Taken together, this further supports a role for disruption of *Nedd4* function in neural crest cells, specifically impacting the morphogenesis of the outflow tract, and not due to loss or misspecification of either the neural crest or second heart field tissues.

Myocardial differentiation is initiated as second heart field cardiac progenitors enter the developing outflow tract, where they upregulate expression of various myocardial markers such as myosin heavy and light chains and other myogenic factors[13]. MF20, which recognises myosin heavy chain, is widely used to assess in vivo differentiation dynamics at the transition zone between the second heart field and outflow tract myocardium[13–16]. While in wildtype embryos the second heart field maintains minimal expression of MF20, *Wnt1-Cre; Nedd4^{fl/fl}* embryos exhibited precocious MF20 expression in the second heart field (Fig. 3G and Supplementary Fig. 4C, D). The enhanced MF20 expression was also observed in ex vivo second heart field explant cultures (Fig. 3H), supporting the inference that loss of *Nedd4* in neural crest cells causes premature differentiation of second heart field cardiac progenitors. Precocious myocardial differentiation is reputed to be one mechanism underpinning defective outflow tract lengthening, due to reducing the pool of cardiac progenitors available for contribution to the growing outflow tract[2]. To assess the effects of premature differentiation on second heart field deployment into the outflow tract, we used EdU to label the highly proliferative second heart field (relative to the lowly-proliferative outflow tract myocardium) at E9.5, and tracked the extent of deployment of EdU +ve cells into the outflow tract 24 h after labelling[17,18]. After a 1 h pulse, EdU labelling of the second heart field was equivalent between genotypes (Fig. 3I). EdU incorporation was terminated 1 h after the EdU pulse by injection of unlabelled thymidine, and EdU+ve cells were chased for 24 h to assess their movement from the second heart field into the

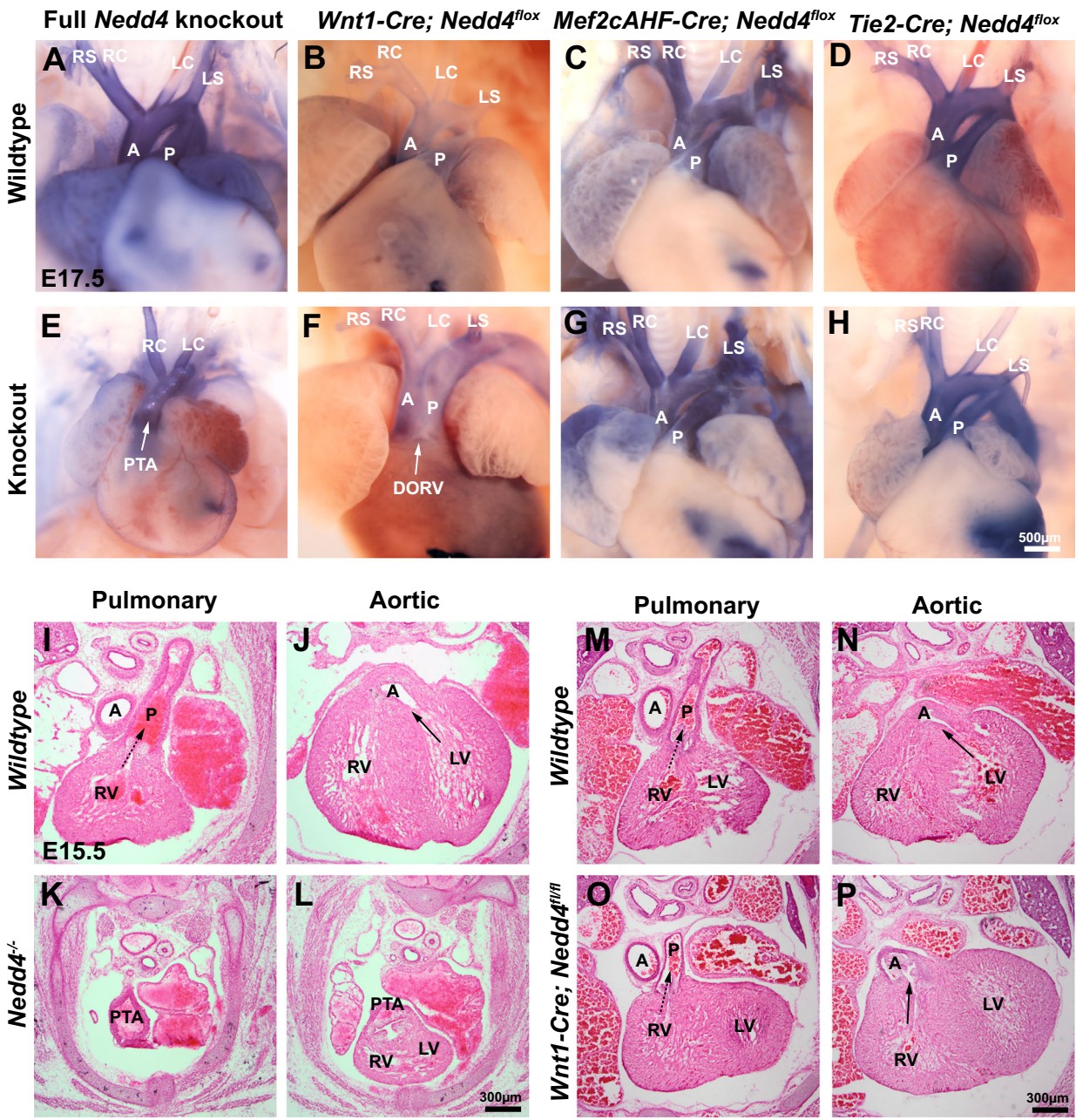

**Q    Distribution of outflow tract defects in *Nedd4⁻/⁻* and *Nedd4^{fl/fl}* crosses (E15.5-E17.5)**

| | Genotype | | | |
|---|---|---|---|---|
| | *Nedd4⁻/⁻* | *Wnt1-Cre; Nedd4^{fl/fl}* | *Mef2cAHF-Cre; Nedd4^{fl/fl}* | *Tie2-Cre; Nedd4^{fl/fl}* |
| Total assessed | 20 | 26 | 8 | 6 |
| All defects | 20 (100%) | 26 (100%) | 0 | 0 |
| Persistent truncus arteriosus | 14 (70%) | 2 (8%) | 0 | 0 |
| Transposition of great arteries | 5 (25%) | 2 (8%) | 0 | 0 |
| Double outlet right ventricle | 1 (5%) | 19 (73%) | 0 | 0 |
| Tetralogy of Fallot (overriding aorta) | 0 | 3 (12%) | 0 | 0 |

**Fig. 1 | *Nedd4* activity in neural crest cells is required for cardiac outflow tract development. A–H** Whole E17.5 hearts from *Nedd4* full- or conditional-knockout mouse strains. Hearts were injected with Evans Blue solution in the right and left ventricles to aid visualisation of the outflow tract and branching arteries. A aorta, P pulmonary artery, RS right subclavian artery, RC right common carotid artery, LC left common carotid artery, LS left subclavian artery, PTA persistent truncus arteriosus, DORV double outlet right ventricle. Representative images from *n* = 3

embryos per genotype. **I–P** Transverse sections through the heart and outflow tract of E15.5 embryos at the level of the pulmonary or aortic valve region from wildtype and *Nedd4⁻/⁻* strains (**I–L**), and from wildtype and *Wnt1-Cre; Nedd4^{fl/fl}* strains (**M–P**), stained with H&E. Dashed arrow indicates ventricle patency with the pulmonary artery, and solid arrow indicates ventricle patency with the aorta. RV right ventricle, LV left ventricle. **Q** Frequency of outflow tract defects in indicated mouse strains, classified by histological staining of tissue sections as in (**I–P**).

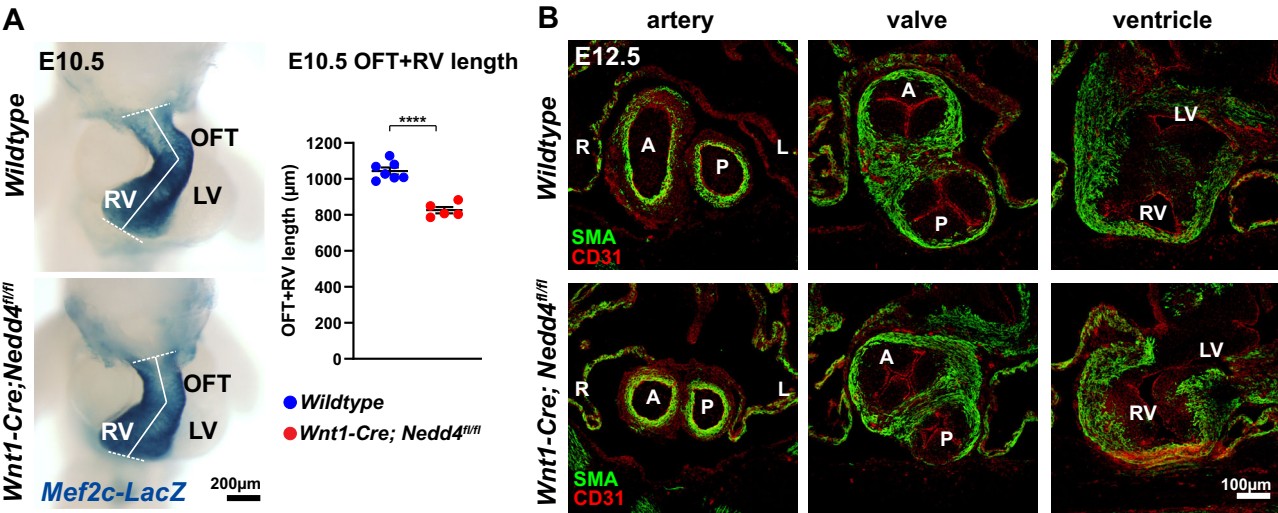

**Fig. 2 | Hearts of *Wnt1-Cre; Nedd4<sup>fl/fl</sup>* embryos exhibit shortened outflow tract and incomplete outflow tract rotation. A** Hearts of *wildtype* and *Wnt1-Cre; Nedd4<sup>fl/fl</sup>* E10.5 embryos showing expression of *Mef2cAHF-LacZ* reporter gene in the right ventricle (RV) and outflow tract (OFT). Right panel: OFT + RV length was measured as the distance along the dashed lines as indicated. LV left ventricle. Data points represent biological replicates from *n* = 7 *wildtype* and *n* = 5 *Wnt1-Cre; Nedd4<sup>fl/fl</sup>* embryos from three independent experiments. Graph represents mean +/− SEM, ****P* = 0.000011 (unpaired *t* test, two-tailed). **B** Coronal sections through the outflow tract region of *wildtype* and *Wnt1-Cre; Nedd4<sup>fl/fl</sup>* E12.5 embryos immunostained for smooth muscle actin (SMA) and CD31. Representative images from *n* = 6 embryos per genotype from six independent experiments. Incomplete clockwise outflow tract rotation is observed in *Wnt1-Cre; Nedd4<sup>fl/fl</sup>* embryos, evidenced by misalignment of the aortic valve over the right ventricle. Valve leaflet defects are also observed. L left, R right, A aortic artery/valve, P pulmonary artery/valve. Source data are provided as a Source Data file.

outflow tract. After a 24 h chase at E10.5, EdU+ve cells had moved a shorter distance into the outflow tract of *Wnt1-Cre; Nedd4<sup>fl/fl</sup>* embryos (Fig. 3J and Supplementary Fig. 5), consistent with impeded deployment of second heart field cells into the outflow tract underpinning outflow tract lengthening defects.

Neural crest cells have been implicated in modulating BMP and FGF signalling in the second heart field to impact outflow tract development[19,20]. Expression of *Bmp4* and *Fgf8*, as well as downstream signalling components phospho-SMAD1/5/9 and phospho-ERK1/2, was not affected in *Wnt1-Cre; Nedd4<sup>fl/fl</sup>* embryos (Supplementary Figs. 3E, F and 6). Hence, we hypothesised that neural crest cells may be regulating the differentiation dynamics of the second heart field via other molecular mechanisms or signalling pathways.

### Neural crest-derived DKK1 modulates Wnt signalling in the second heart field

To investigate how loss of *Nedd4* in neural crest cells causes premature second heart field differentiation, we used laser capture microdissection to isolate cells from a defined region of the anterior second field, as well as neural crest cells and pharyngeal endoderm in the immediate vicinity, from wild-type and *Wnt1-Cre; Nedd4<sup>fl/fl</sup>* embryos for transcriptome profiling (Fig. 4A and Supplementary Dataset 1). 897 differentially expressed genes (DEGs) with *P* < 0.05 were identified (Fig. 4B). Gene ontology enrichment analysis of the top 30 DEGs (Fig. 4C) revealed Wnt signalling as the top signalling pathway with differential gene expression (Supplementary Fig. 7A, B). *Dkk1*, the gene encoding a secreted Wnt signalling antagonist, was the top DEG identified (Fig. 4C). We next performed immunostaining and in situ hybridisation to investigate cell-type specificity of gene expression. Immunostaining revealed DKK1 expression was localised exclusively to neural crest cells (that co-express the neural crest marker AP2α), with no expression in other cell lineages such as the second heart field or pharyngeal endoderm (Fig. 4D). Importantly, DKK1 was only expressed in cardiac neural crest cells that had reached the peri-second heart field region in their migration trajectory, and not in earlier post-delaminating or migrating neural crest cells (Supplementary Fig. 8). In situ hybridisation, in parallel with immunostaining, validated the enhanced *Dkk1* expression and enriched DKK1 abundance in cardiac neural crest cells in *Wnt1-Cre; Nedd4<sup>fl/fl</sup>* embryos (Fig. 4E–G). Given the function of DKK1 as a canonical Wnt signalling inhibitor, and the proximity of cardiac neural crest cells to the anterior second heart field, we hypothesised that neural crest-derived DKK1 may antagonise Wnt signalling in the adjacent second heart field. Employing β-catenin protein stabilisation and nuclear translocation as a readout of canonical Wnt signalling activity[21], immunostaining for active forms of β-catenin (non-pS45, ABC (non-pS37/T41) and pY489[22,23]) revealed a specific reduction in nuclear localised staining intensity in the anterior second heart field of *Wnt1-Cre; Nedd4<sup>fl/fl</sup>* embryos (Fig. 4H and Supplementary Fig. 9A–E). Furthermore, we observed a reduction in Pitx2 expression, a canonical Wnt signalling target gene[24] in the second heart field of *Wnt1-Cre; Nedd4<sup>fl/fl</sup>* embryos (Supplementary Fig. 9G, H). As well as signalling functions, β-catenin is also involved in regulating cell-adhesion via interactions with cadherins and the actin cytoskeleton[25]. Recent work has demonstrated requisite roles for cell-adhesion, polarity, and epithelial-like properties of the second heart field in outflow tract development[13,26]. Immunostaining for markers of cell-adhesion, polarity, and ECM components did not reveal any changes of these cellular attributes in the second heart field region of *Wnt1-Cre; Nedd4<sup>fl/fl</sup>* embryos (Supplementary Fig. 10). Taken together, our data support a role for neural crest-derived DKK1 in modulating canonical Wnt signalling activity in the second heart field.

To assess canonical Wnt signalling activity in *Wnt1-Cre; Nedd4<sup>fl/fl</sup>* embryos, we intersected our laser-capture tissue mRNAseq dataset with a β-catenin ChIPseq dataset[27] to analyse expression of canonical Wnt target genes. We identified 179 differentially expressed Wnt target genes, with 49 upregulated and 130 downregulated (Fig. 4I). To validate and determine cell-type specificity of expression, we performed in situ hybridisation for ten candidate genes on tissue sections of the outflow tract region from E9.5 *wildtype* and *Wnt1-Cre; Nedd4<sup>fl/fl</sup>* embryos. Downregulated Wnt target genes, e.g., *Axin2, SP5, Pitx2c, Fam49a, Mical2*, were localised to the second heart field (Fig. 4K and Supplementary Fig. 11), consistent with the reduction in nuclear β-catenin in the second heart field. Of note, upregulated expression of Wnt target genes was observed in the cardiac neural crest cells (Fig. 4J),

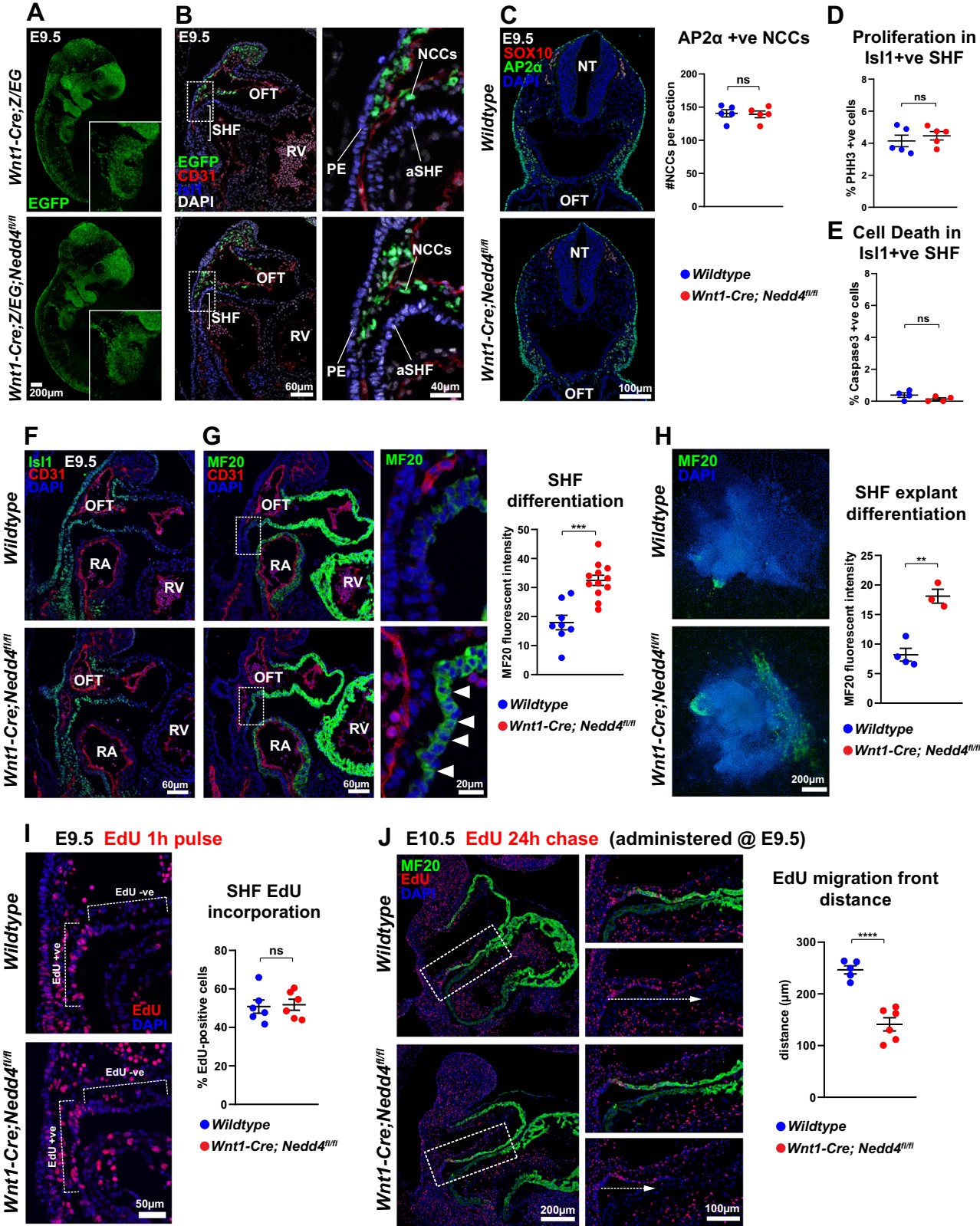

suggesting there may be a cell-autonomous role for *Nedd4* in regulating Wnt activity in neural crest cells.

**Disruption of Wnt signalling activity underpins outflow defects**

To ascertain that disruption of Wnt signalling activity causes outflow tract defects in *Wnt1-Cre; Nedd4^{fl/fl}* embryos, we manipulated canonical Wnt signalling in vivo by *in utero* treatment with small molecule activators (CHIR99021) and inhibitors (iCRT3) of Wnt signalling over the developmental time-period when neural crest cells interact with the second heart field (Fig. 5A). Treatment with Wnt agonist CHIR99021, which may enhance Wnt signalling activity, did not rescue the outflow tract defects in *Wnt1-Cre; Nedd4^{fl/fl}* embryos by E13.5 (Fig. 5B, C and Supplementary Fig. 12B), suggesting that escalating Wnt signalling activity could not counteract the effect of elevated DKK1

**Fig. 3 | *Wnt1-Cre; Nedd4^fl/fl* embryos display precocious differentiation and defective deployment of anterior second heart field derivatives. A** Whole E9.5 *Wnt1-Cre; Z/EG* and *Wnt1-Cre; Z/EG; Nedd4^fl/fl* embryos immunostained for EGFP, which labels neural crest cells and derivatives. The inset image shows the migratory cardiac neural crest cells in the 3rd, 4th, and 6th pharyngeal arches. Representative images from *n* = 3 embryos per genotype from three independent experiments. **B** Sagittal sections through the outflow tract of E9.5 *Wnt1-Cre; Z/EG* and *Wnt1-Cre; Z/ EG; Nedd4^fl/fl* embryos immunostained for EGFP, CD31, and Isl1, revealing localisation of neural crest cells (NCCs) in close proximity to the anterior second heart field (SHF), and outflow tract (OFT). RV right ventricle, PE pharyngeal endoderm. Representative images from *n* = 8 embryos per genotype from eight independent experiments. **C** Transverse sections of the post-otic region of E9.5 *wildtype* and *Wnt1-Cre; Nedd4^fl/fl* embryos immunostained for NCC markers AP2α and SOX10. AP2α is also expressed in the embryonic epithelium; these cells were not included in NCC quantitation. NT neural tube. The graph represents the total number of AP2α-positive NCCs per section, with *n* = 5 embryos per genotype assessed in 5 individual experiments. NCC numbers are equivalent between genotypes. Data points represent mean +/– SEM; ns, not significant, unpaired two-tailed *t* test. **D** Proliferation of SHF cells at E9.5 assessed by immunostaining of sections for phospho-Histone H3 (PHH3) and Isl1; double-positive cells are expressed as a percentage of total Isl1-positive cells. Graph represents values from *n* = 5 embryos per genotype assessed in five individual experiments; mean +/– SEM; ns, not significant, unpaired two-tailed *t* test. **E** Cell death in the SHF at E9.5 assessed by immunostaining of sections for cleaved-caspase-3 and Isl1; double-positive cells are expressed as a percentage of total Isl1-positive cells. Graph represents values from *n* = 5 embryos per genotype assessed in 5 individual experiments; mean +/– SEM; ns, not significant, unpaired two-tailed *t* test. **F** Sagittal sections of E9.5 *wildtype* and *Wnt1-Cre; Nedd4^fl/fl* embryos immunostained for Isl1 and CD31. Representative

images from *n* = 5 embryos per genotype from five independent experiments. **G** Adjacent serial section to F immunostained for MF20 and CD31. The inset shows a magnified view of anterior SHF, with arrowheads indicating precocious SHF differentiation in the mutant embryo. Right panel: Quantification of MF20 mean fluorescent intensity in the anterior SHF. Data points represent biological replicates from *n* = 8 *wildtype* and *n* = 12 *Wnt1-Cre; Nedd4^fl/fl* embryos from seven independent experiments. Graph represents mean +/– SEM, ***P* = 0.000108 (unpaired *t* test, two-tailed). **H** Second heart field explant of *wildtype* and *Wnt1-Cre; Nedd4^fl/fl* embryos, immunostained for MF20, with quantified mean fluorescent intensity. Data points represent biological replicates from *n* = 4 *wildtype* and *n* = 3 *Wnt1-Cre; Nedd4^fl/fl* explants from two independent experiments. Graph represents mean +/– SEM, ***P* = 0.001682 (unpaired *t* test, two-tailed). **I** Sagittal sections through the outflow tract of E9.5 *wildtype* and *Wnt1-Cre; Nedd4^fl/fl* embryos treated with EdU 1 h prior to collection. Fluorescent staining for EdU incorporation highlights low levels of EdU incorporation in the OFT, but high levels of incorporation in the highly proliferative second heart field. After 1 h of treatment, labelling of EdU +ve cells in the SHF is equivalent between *wildtype* and *Wnt1-Cre; Nedd4^fl/fl* embryos. Data points represent biological replicates from *n* = 6 *wildtype* and *n* = 6 *Wnt1-Cre; Nedd4^fl/fl* embryos from three independent experiments. Graph represents mean +/– SEM, ns, not significant (unpaired *t* test, two-tailed). **J** Sagittal sections through the outflow tract of E10.5 *wildtype* and *Wnt1-Cre; Nedd4^fl/fl* embryos treated with 1 h EdU pulse at E9.5, followed by 24 h chase. EdU marks cells that were labelled at E9.5 and have now been deployed from the SHF into the OFT. The migration distance of EdU+ cells was measured along the dashed arrow, and the distance quantified (right panel). Data points represent biological replicates from *n* = 5 *wildtype* and *n* = 6 *Wnt1-Cre; Nedd4^fl/fl* embryos from three independent experiments. Graph represents mean +/– SEM, ****P* = 0.000094 (unpaired *t* test, two-tailed). Source data are provided as a Source Data file.

activity. However, when Wnt signalling was blocked with the chemical inhibitor iCRT3, *Wnt1-Cre; Nedd4^fl/+* heterozygous embryos, which do not normally exhibit structural heart defects, now exhibited outflow tract rotation defects (Fig. 5B, C and Supplementary Fig. 12C). Thus, loss of one copy of *Nedd4* in neural crest cells sensitises the phenotypic response to disruption of Wnt signalling, and points to genetic interaction between *Nedd4* and the Wnt signalling pathway.

To determine if the outflow tract rotation deficiency in iCRT3-treated *Wnt1-Cre; Nedd4^fl/+* heterozygous embryos is caused by premature second heart field differentiation, we examined the differentiation of second heart field tissue in E9.5 + 6 h iCRT3-treated embryos by MF20 immunostaining (Fig. 5D). iCRT3-treated *Wnt1-Cre; Nedd4^fl/+* heterozygous embryos exhibited precocious second heart field differentiation, like that in *Wnt1-Cre; Nedd4^fl/fl* knockout embryos (Fig. 5E, F and Supplementary Fig. 4E, F). Furthermore, examination of Wnt signalling activity using β-catenin pY489 and non-pS45 immunostaining revealed reduced nuclear-localised β-catenin in the second heart field of iCRT3-treated *Wnt1-Cre; Nedd4^fl/+* heterozygous embryos, indistinguishable from that of *Wnt1-Cre; Nedd4^fl/fl* knockout embryos (Supplementary Fig. 13). These findings suggest that a reduced level of canonical Wnt signalling underpins premature second heart field differentiation, and is causative of outflow tract rotation deficiency.

To determine if ectopic *Dkk1* overexpression is a contributing factor to outflow tract defects, we generated compound *Wnt1-Cre; Nedd4^fl/fl; Dkk1^+/-* embryos in which *Dkk1* expression was reduced in the *Wnt1-Cre; Nedd4^fl/fl* background, to test for genetic rescue of outflow tract defects. Compared to *Wnt1-Cre; Nedd4^fl/fl* embryos, *Wnt1-Cre; Nedd4^fl/fl; Dkk1^+/-* embryos exhibited improvement in outflow tract development at E13.5 (Fig. 5G, H and Supplementary Fig. 12D), including restoration of aortic valve diameter (Fig. 5I and Supplementary Fig. 14A) partially rescued pulmonary valve diameter (Fig. 5J and Supplementary Fig. 14B) and partial rescue of outflow tract tissue mass (Fig. 5K and Supplementary Fig. 14C). While artery-ventricle alignment was not completely rescued in *Wnt1-Cre; Nedd4^fl/fl; Dkk1^+/-* embryos, there was a greater extent of outflow tract rotation (Fig. 5L and Supplementary Fig. 12D). Taken together, the enhanced outflow tract development and rotation suggest that ectopic *Dkk1*

overexpression is one major factor underpinning defective outflow tract development in *Wnt1-Cre; Nedd4^fl/fl* embryos.

## DKK1 is a substrate for NEDD4-mediated ubiquitination

Expression of DKK1, at both transcript and protein levels, was elevated in neural crest cells with loss of *Nedd4* function (Fig. 4). Given NEDD4 is an E3 ubiquitin ligase with important roles in regulating protein abundance, we investigated if DKK1 may be a ubiquitinated substrate of NEDD4. Computational analysis on the UbiBrowser bioinformatic platform[28] predicts DKK1 can be ubiquitinated, with NEDD4 predicted to be the targeting E3 ligase. Using in vitro ubiquitination assays, we demonstrated that wildtype NEDD4 indeed ubiquitinates DKK1, while an inactive version of NEDD4 in which the catalytic cysteine is mutated (NEDD4 CS) is unable to ubiquitinate DKK1 (Fig. 6A and Supplementary Fig. 15A). Moreover, NEDD4 specifically ubiquitinates DKK1, but not other DKK family members such as DKK2 (Fig. 6B). This ubiquitination likely targets DKK1 for protein degradation, as overexpression of NEDD4 WT in HEK293T cells reduces the steady-state levels of DKK1-Myc, whereas overexpression of NEDD4 CS did not affect DKK1-Myc levels (Fig. 6C). Overexpression of DKK1-GFP and FLAG-NEDD4 in HeLa cells revealed an inverse correlation of fluorescence intensity that measured protein abundance between DKK1 and NEDD4 protein, but correlation was observed between DKK1 and NEDD4 CS (Fig. 6D and Supplementary Fig. 16A). Expression of a GFP-only construct with NEDD4 WT and NEDD4 CS revealed no inverse correlation (Fig. 6E), suggesting that DKK1 is the substrate targeted by NEDD4. Furthermore, high-magnification imaging of cardiac neural crest cells in vivo revealed co-localisation of NEDD4 and DKK1 in wild-type neural crest cells, with greatly increased DKK1 fluorescent intensity and abundance in *Wnt1-Cre; Nedd4^fl/fl* cardiac neural crest cells (Fig. 6F), consistent with the inverse correlation of fluorescence intensity that was observed in the in vitro HeLa cell assay. Interestingly, NEDD4 exhibited a particularly enhanced expression and punctate subcellular localisation specifically in cardiac neural crest cells (Supplementary Fig. 1E and Fig. 6F), suggesting a potentially unique function for NEDD4 in this cell type. Taken together, the increased levels of DKK1 in the neural crest cells of *Wnt1-Cre; Nedd4^fl/fl* embryos can be accounted for in part by the

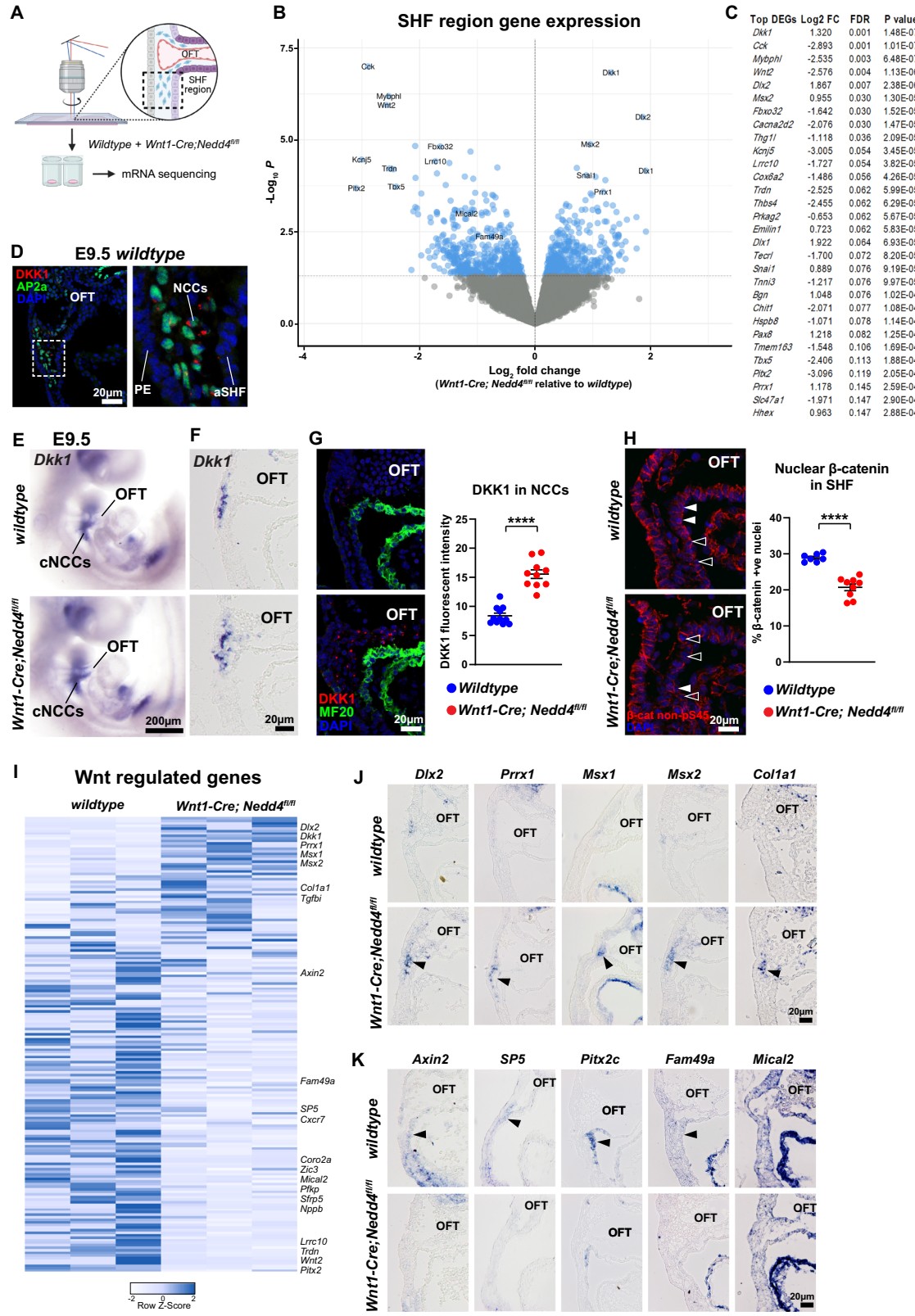

**B** SHF region gene expression

**C**

| Top DEGs | Log2 FC | FDR | P value |
|---|---|---|---|
| *Dkk1* | 1.320 | 0.001 | 1.48E-07 |
| *Cck* | -2.893 | 0.001 | 1.01E-07 |
| *Mybphl* | -2.535 | 0.003 | 6.48E-07 |
| *Wnt2* | -2.576 | 0.004 | 1.13E-06 |
| *Dlx2* | 1.867 | 0.007 | 2.38E-06 |
| *Msx2* | 0.955 | 0.030 | 1.30E-05 |
| *Fbxo32* | -1.642 | 0.030 | 1.52E-05 |
| *Cacna2d2* | -2.076 | 0.030 | 1.47E-05 |
| *Thg1l* | -1.118 | 0.036 | 2.09E-05 |
| *Kcnj5* | -3.005 | 0.054 | 3.45E-05 |
| *Lrrc10* | -1.727 | 0.054 | 3.82E-05 |
| *Cox6a2* | -1.486 | 0.056 | 4.26E-05 |
| *Trdn* | -2.525 | 0.062 | 5.99E-05 |
| *Thbs4* | -2.455 | 0.062 | 6.29E-05 |
| *Prkag2* | -0.653 | 0.062 | 5.67E-05 |
| *Emilin1* | 0.723 | 0.062 | 5.83E-05 |
| *Dlx1* | 1.922 | 0.064 | 6.93E-05 |
| *Tecrl* | -1.700 | 0.072 | 8.20E-05 |
| *Snai1* | 0.889 | 0.076 | 9.19E-05 |
| *Tnni3* | -1.217 | 0.076 | 9.97E-05 |
| *Bgn* | 1.048 | 0.076 | 1.02E-04 |
| *Chit1* | -2.071 | 0.077 | 1.08E-04 |
| *Hspb8* | -1.071 | 0.078 | 1.14E-04 |
| *Pax8* | 1.218 | 0.082 | 1.25E-04 |
| *Tmem163* | -1.548 | 0.106 | 1.69E-04 |
| *Tbx5* | -2.406 | 0.113 | 1.88E-04 |
| *Pitx2* | -3.096 | 0.119 | 2.05E-04 |
| *Prrx1* | 1.178 | 0.145 | 2.59E-04 |
| *Slc47a1* | -1.971 | 0.147 | 2.90E-04 |
| *Hhex* | 0.963 | 0.147 | 2.88E-04 |

absence of NEDD4-mediated ubiquitination and consequent degradation of DKK1.

## NEDD4 is clinically implicated in congenital heart defects

We identified a homozygous missense variant in *NEDD4* in a child with Tetralogy of Fallot. The child and parents were recruited and assessed as part of an Australian congenital heart disease genome sequencing study[29] (now extended to *n* = 363 trios and singletons). Analysis of genome sequencing data from the trio revealed the recessive inheritance of a *NEDD4* variant by the proband from parents carrying one allele each (Fig. 7A). The variant NEDD4:c.2297 A > G:p.K766R (GenBank:NM_006154.4) changes the reference lysine to an arginine, which is predicted to be damaging by in silico metrics CADD and PolyPhen (CADD: 25, PolyPhen 0.995). No predicted damaging variants were

**Fig. 4 | Laser capture mRNA sequencing reveals disrupted Wnt signalling in *Wnt1-Cre; Nedd4^{fl/fl}* embryos. A** Diagram indicating tissue region (dashed box) that was excised by laser capture for mRNA sequencing, from $n = 3$ *wildtype* and *Wnt1-Cre; Nedd4^{fl/fl}* embryos. Created in BioRender. Schwarz, Q. (2026) https://BioRender.com/g0bv7cx. **B** Volcano plot representing differentially expressed genes (DEGs) in *Wnt1-Cre; Nedd4^{fl/fl}* tissue samples relative to *wildtype*. **C** List of top 30 DEGs identified from laser capture mRNA sequencing. **D** Sagittal section through the outflow tract of an E9.5 *wildtype* embryo immunostained for DKK1 and AP2α, which marks the nucleus of cardiac neural crest cells (cNCCs). Inset: higher magnification demonstrates DKK1 is expressed in neural crest cells, but not in anterior second heart field (aSHF) or pharyngeal endoderm (PE). Representative images from $n = 3$ independent experiments. **E** Wholemount in situ hybridisation of E9.5 *wildtype* and *Wnt1-Cre; Nedd4^{fl/fl}* embryos, showing *Dkk1* expression in cardiac NCCs. Representative images from $n = 5$ embryos per genotype from 5 independent experiments. **F** In situ hybridisation of E9.5 *wildtype* and *Wnt1-Cre; Nedd4^{fl/fl}* embryo sagittal sections showing *Dkk1* expression in cardiac NCCs. OFT outflow tract. Representative images from $n = 5$ embryos per genotype from five independent experiments. **G** Sagittal sections of E9.5 *wildtype* and *Wnt1-Cre; Nedd4^{fl/fl}* embryos immunostained for DKK1 and MF20, highlighting increased expression of DKK1 in

NCCs in *Wnt1-Cre; Nedd4^{fl/fl}* embryos, and quantified fluorescence intensity. Data points represent biological replicates from $n = 11$ *wildtype* and $n = 10$ *Wnt1-Cre; Nedd4^{fl/fl}* embryos from seven independent experiments. Graph represents mean +/− SEM, ****$P = 0.0000009$ (unpaired *t* test, two-tailed). **H** Sagittal sections of E9.5 *wildtype* and *Wnt1-Cre; Nedd4^{fl/fl}* embryos immunostained for β-catenin non-pS45. Solid arrowheads mark examples of nuclear localised β-catenin, while outlined arrowheads mark examples of no nuclear β-catenin accumulation. Nuclear β-catenin staining in the SHF is quantified. Data points represent biological replicates from $n = 7$ *wildtype* and $n = 9$ *Wnt1-Cre; Nedd4^{fl/fl}* embryos from 6 independent experiments. Graph represents mean +/− SEM, ****$P = 0.000005$ (unpaired *t* test, two-tailed). **I** Heatmap of differentially expressed Wnt-regulated genes. **J** In situ hybridisation of E9.5 *wildtype* and *Wnt1-Cre; Nedd4^{fl/fl}* embryo sagittal sections, showing increased expression of Wnt-regulated genes in *Wnt1-Cre; Nedd4^{fl/fl}* embryos. Arrowhead indicates expression in neural crest cells. **K** In situ hybridisation of *Wnt1-Cre; Nedd4^{fl/fl}* embryos showing diminished expression of Wnt-regulated genes. Arrowhead indicates expression in the second heart field tissue. (**J, K,** representative images from $n = 3$ embryos per genotype from three independent experiments). Source data are provided as a Source Data file.

identified in other genes presently known to cause congenital heart disease, making the *NEDD4* variant a compelling candidate for causing disease in this family. The parents were not consanguineous. While the family also reported congenital heart disease in distant relatives on the maternal side of the family, no clinical data were available. Furthermore, an additional *NEDD4* missense variant was recently documented in another individual with Tetralogy of Fallot in an independent study[30], strengthening disease links to this gene.

The K766R amino acid change is located in the HECT domain of NEDD4, which is responsible for ubiquitin ligase activity (Fig. 7B). Given that NEDD4 can ubiquitinate DKK1, in vitro assays to assess the effect of K766R amino acid substitution on NEDD4 function revealed NEDD4 K766R had reduced capacity to ubiquitinate DKK1 (Fig. 7C and Supplementary Fig. 15B). Furthermore, analysis of fluorescence intensity revealed no correlation between DKK1 and NEDD4 K766R protein levels (Fig. 7D and Supplementary Fig. 16B), confirming NEDD4 K766R is a loss-of-function variant for ubiquitination of DKK1.

To investigate the in vivo consequences of NEDD4 K766R substitution, we generated a CRISPR-engineered *Nedd4^{K766R/K766R}* mouse model. Homozygous embryos of this mutant mouse strain were present at expected mendelian ratios at all developmental stages and phenotypically normal (Fig. 7E). Examination of hearts in E15.5 to E17.5 *Nedd4^{K766R/K766R}* homozygous embryos revealed a range of developmental cardiac defects (Fig. 7F). While artery-ventricle alignment defects such as overriding aorta, which is a clinical feature of Tetralogy of Fallot, were not present, there were defects consistent with deficiency of right ventricular development. This included a thin ventricular wall which was restricted to the right ventricle, often with a rugged appearance (Fig. 7G, H) suggestive of defective right ventricular compaction, a disorganised interventricular septum, and ventricular septal defects (Fig. 7G). Hence, *Nedd4^{K766R/K766R}* embryos exhibit heart defects, including some features of Tetralogy of Fallot, such as right ventricular wall and interventricular septal defects, implicating the NEDD4 K766R variant as a cause of congenital heart disease.

Taken together, NEDD4 regulation of DKK1 in neural crest cells unveils a novel mechanistic paradigm in heart development, which when disrupted leads to heart defects in mice and humans.

## Discussion

Wnt signalling plays critical roles in multiple aspects of heart development, where it has stage-dependent positive and negative effects on progenitor cell specification, maintenance, and differentiation (reviewed in refs. 31,32). Canonical Wnt/β-catenin signalling is required for initial mesoderm induction in the gastrulating embryo; however, it

is then downregulated to promote cardiac precursor specification, which concomitantly is supported by activation of non-canonical Wnt signalling[33,34]. As heart development progresses, cells of the second heart field require canonical Wnt/β-catenin signalling to maintain cells in a proliferative and undifferentiated state. As second heart field progenitors move into the outflow tract, canonical Wnt signalling is gradually downregulated, arresting cell proliferation and initiating myocardial differentiation, in which non-canonical Wnt signalling plays an important role. Hence, the temporal balance of activation and inhibition of Wnt signalling is essential for orchestrating the proliferation and differentiation dynamics of cardiac progenitors during heart development.

Conditional knockout and constitutive activation of β-catenin in specific cardiac cell lineages support a role for the dynamic regulation of canonical Wnt signalling in heart development. Knockout of β-catenin in the *Islet1-Cre* or *SM22α-Cre* lineages results in failure of the right ventricle to form, while overactivation of β-catenin causes right ventricular enlargement[35,36]. In other lineages (*Mef2cAHF-Cre* and *MesP1-Cre*), both loss- and gain-of-function of β-catenin are detrimental to right ventricle formation[37,38]. Hence, the precise dosage and timing of Wnt/β-catenin signalling is critical. Inactivation of β-catenin target genes in the second heart field also supports a role for canonical Wnt signalling in outflow tract development. For example, conditional knockout of *Pitx2* in *Mef2cAHF-Cre* and *Islet1-Cre* lineages causes artery-ventricle alignment defects including double outlet right ventricle and transposition of the great arteries[39], and is consistent with the phenotype of *Wnt1-Cre; Nedd4^{fl/fl}* embryos which exhibit downregulation of *Pitx2c* in the second heart field.

While Wnt signalling is known for its role in second heart field development, the source of Wnt ligands has remained unclear. A recent study suggests that Wnt2 secreted from the first heart field can promote second heart field proliferation at early stages of heart development (E8.5)[40]. Whether Wnt2 is the key Wnt ligand responsible for maintaining the second heart field in a proliferative state at later stages (E9.5) during the critical time period of outflow tract lengthening remains to be determined. Nevertheless, as anterior second heart field cells move into the outflow tract, canonical Wnt signalling is downregulated to promote myocardial differentiation. This reduction in Wnt signalling may be a consequence of these cells moving away from a localised source of Wnt morphogen as they enter the outflow tract. Alternatively, we propose that cardiac neural crest cells can act as a critical rheostat of Wnt signalling as second heart field cells transition into the outflow tract. Neural crest cells are spatially positioned in this transition zone; providing a source of DKK1 to fine-tune Wnt signals and promote coordinated second heart field differentiation.

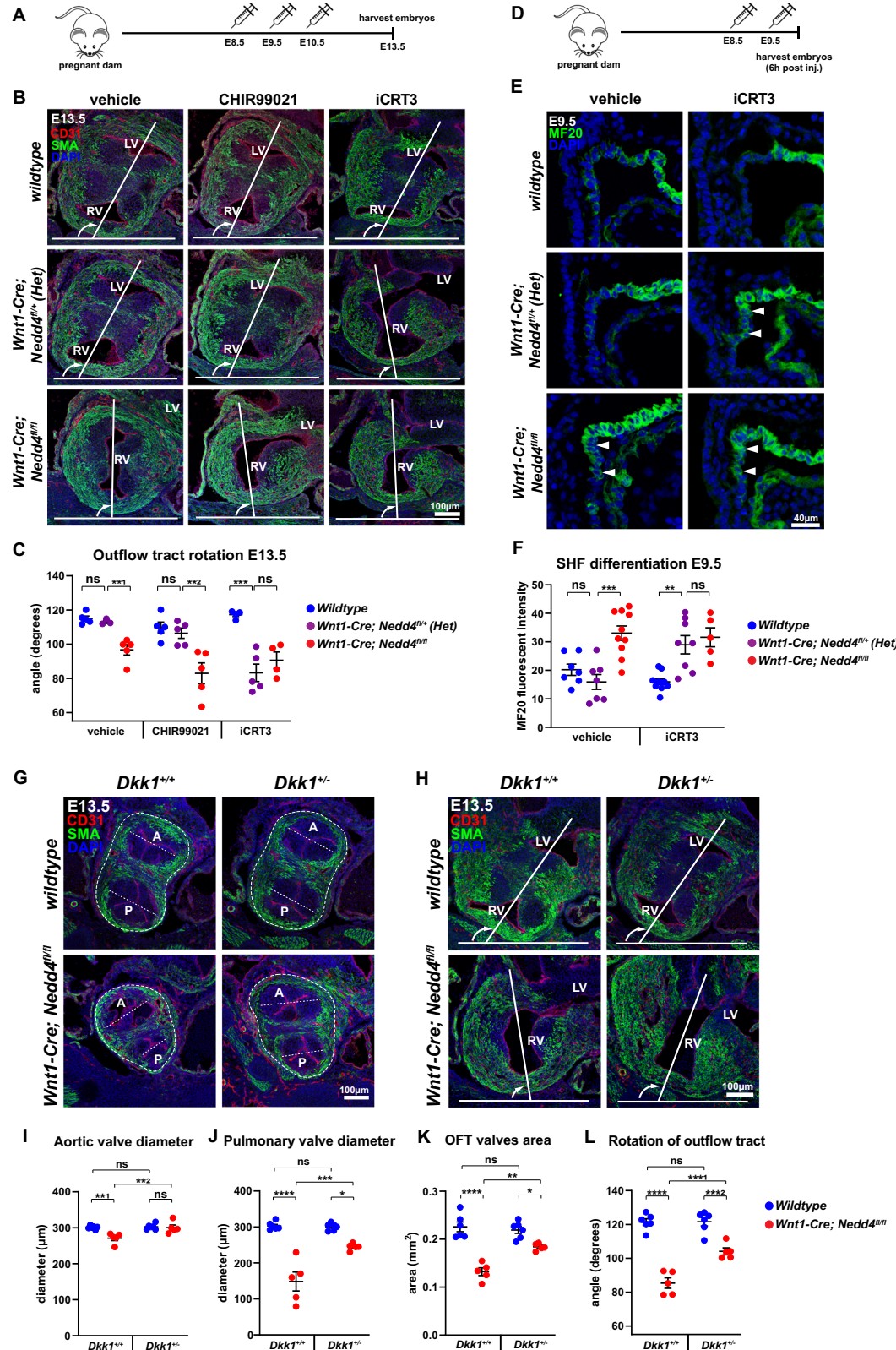

Neural crest cells have previously been implicated in second heart field development. Early studies in avian models demonstrated that surgical ablation of the premigratory cardiac neural crest caused an array of conotruncal heart defects[4], underpinned by failed addition of second heart field cells to the lengthening outflow tract[6,7]. Furthermore, these studies showed loss of cardiac neural crest cells led to increased FGF signalling and excessive proliferation in the second heart field, suggesting cardiac neural crest act to modulate FGF signalling to promote appropriate differentiation of the outflow tract myocardium[5,41,42]. BMP signalling antagonises the pro-proliferative effects of FGF signalling to promote myocardial differentiation, and neural crest cells have been shown to be essential for mediating the effects of BMP on cardiomyocyte differentiation[19,20]. Disruption of downstream BMP signalling components in neural crest cells also

**Fig. 5 | Reduced canonical Wnt signalling underpins precocious second heart field differentiation. A** Protocol for drug administration at E8.5, E9.5, and E10.5, and sample collection at E13.5. **B** Coronal sections through the outflow tract region of E13.5 *wildtype*, *Wnt1-Cre; Nedd4^fl/+* and *Wnt1-Cre; Nedd4^fl/fl* embryos immunostained for smooth muscle actin (SMA) and CD31. RV and LV indicate the positions of the right ventricle and left ventricle, respectively. Outflow tract rotation was measured as the degree of clockwise rotation, indicated from the curved arrow from a perpendicular reference plane (the transverse plane of the embryo). **C** Quantitation of outflow tract rotation. *Wnt1-Cre; Nedd4^fl/fl* embryos exhibit deficient rotation in all treatment conditions, while *Wnt1-Cre; Nedd4^fl/+* embryos also exhibit deficient rotation only when treated with the canonical Wnt signalling inhibitor iCRT3, and are indistinguishable from *Wnt1-Cre; Nedd4^fl/fl* embryos. Data points represent biological replicates n = 14 *wildtype*, n = 13 *Wnt1-Cre; Nedd4^fl/+* and n = 14 *Wnt1-Cre; Nedd4^fl/fl* embryos from 9 independent experiments. Graph represents mean +/− SEM. **\*¹** P = 0.0024; **\*²** P = 0.0063; **\*\*\*** P = 0.0006; ns, not significant (one-way ANOVA multiple comparisons). **D** Protocol for drug administration at E8.5 and E9.5, and sample collection 6 h after last treatment. **E** Sagittal sections through the outflow tract region of E9.5 *wildtype*, *Wnt1-Cre; Nedd4^fl/+* and *Wnt1-Cre; Nedd4^fl/fl* embryos immunostained for MF20. Arrowheads indicate precocious SHF differentiation. **F** Quantitation of MF20 fluorescent intensity in the SHF. *Wnt1-Cre; Nedd4^fl/+* embryos exhibit precocious SHF differentiation when treated with iCRT3, and are indistinguishable from *Wnt1-Cre; Nedd4^fl/fl* embryos. Data points represent biological replicates n = 17 *wildtype*, n = 15 *Wnt1-Cre; Nedd4^fl/+* and n = 15 *Wnt1-Cre;*

*Nedd4^fl/fl* embryos from 13 independent experiments. Graph represents mean +/− SEM. **\*\*\*** P = 0.0001; **\*\*** P = 0.0035; ns, not significant (two-way ANOVA multiple comparisons). **G** Coronal sections through the outflow tract valves of E13.5 *wildtype* and *Wnt1-Cre; Nedd4^fl/fl* embryos crossed to *Dkk1^+/−* and immunostained for SMA and CD31. A aortic valve, P pulmonary valve. Dashed lines represent the valvular diameter and OFT tissue area measured. **H** Coronal sections through the outflow tract region of E13.5 *wildtype* and *Wnt1-Cre; Nedd4^fl/fl* embryos crossed to *Dkk1^+/−* and immunostained for SMA and CD31. Outflow tract rotation was measured as in (**B**). **I–K** Quantitation of aortic and pulmonary valve diameter, and outflow tract area as indicated in (**G**). These measurements are either fully or partially restored in *Wnt1-Cre; Nedd4^fl/fl; Dkk1^+/−* embryos. Data points represent biological replicates n = 6 *Dkk1^+/+,wildtype*; n = 5 *Dkk1^+/+,Wnt1-Cre; Nedd4^fl/fl*; n = 6 *Dkk1^+/−,wildtype*; and n = 5 *Dkk1^+/−,Wnt1-Cre; Nedd4^fl/fl* embryos from five independent experiments. Graphs represent mean +/− SEM. **I \*¹** P = 0.0019; **\*²** P = 0.0035 (**J**) **\*\*\*\*** P = 0.0000009; **\*\*\*** P = 0.00032; **\*** P = 0.019 (**K**) **\*\*\*\*** P = 0.0000009; **\*\*** P = 0.0016; **\*** P = 0.026; ns, not significant (two-way ANOVA multiple comparisons). **L** Quantitation of outflow tract rotation from images in (**H**). Rotation is partially restored in *Wnt1-Cre; Nedd4^fl/fl; Dkk1^+/−* embryos. Data points represent biological replicates n = 6 *Dkk1^+/+,wildtype*; n = 5 *Dkk1^+/+,Wnt1-Cre; Nedd4^fl/fl*; n = 6 *Dkk1^+/−,wildtype*; and n = 5 *Dkk1^+/−,Wnt1-Cre; Nedd4^fl/fl* embryos from five independent experiments. Graphs represent mean +/− SEM. **\*\*\*\*** P = 0.0000009; **\*\*\*¹** P = 0.00029; **\*\*²** P = 0.00038; ns, not significant. Source data are provided as a Source Data file.

---

causes abnormal second heart field differentiation and outflow tract defects[43,44]. How neural crest cells mediate BMP-induced second heart field differentiation is unknown, however an attractive hypothesis is that BMP signalling may induce neural crest cells to secrete another factor which in turn suppresses FGF signalling and/or promotes differentiation. Wnt/β-catenin signalling also feeds into these complex signalling interactions, with Wnt gain- and loss-of-function also affecting BMP and FGF signalling outcomes in the second heart field[36–38]. As no disruption of FGF and BMP signalling, or changes in proliferation of second heart field progenitors were observed in *Wnt1-Cre; Nedd4^fl/fl* embryos, this may suggest neural crest-derived factors act downstream of these signalling interactions in this model to regulate myocardial differentiation of the second heart field. Hence, FGF and BMP signals may be essential to prime the second heart field to appropriately respond to differentiation signals, with neural crest cells acting as a critical source of DKK1 to modulate Wnt signalling and actively initiate myocardial differentiation.

DKK1 is ideally located to act as a rheostat of Wnt signalling, providing precise spatial and temporal control of canonical Wnt signalling activity in the second heart field. This dynamic regulation is essential for maintaining a critical balance of second heart field progenitor maintenance versus differentiation, to foster appropriate outflow tract elongation. Our study has unveiled the expression of DKK1 specifically in cardiac neural crest cells, providing new mechanistic data to explain how neural crest cells regulate second heart field myocardial differentiation. Enhanced DKK1 activity induces cardiac differentiation in embryonic stem cell models[35,45], and in Xenopus embryos[46], which is consistent with the increased myocardial differentiation observed in the second heart field of *Wnt1-Cre; Nedd4^fl/fl* embryos. While *Dkk1* knockout mouse embryos have no reported heart defects[47], *Dkk1* and *Dkk2* double knockout embryos exhibit mild heart defects, including ventricular septal defects and myocardial/epicardial hyperplasia[48]. Findings of our present study are supportive of ectopic gain-of-function of DKK1, rather than loss-of-function, as being causative of developmental defects of the heart. The impact of widespread ectopic gain-of-function of *Dkk1* on heart development remains to be investigated.

Our study reports the ubiquitin modification of the DKK1 protein as an additional level of post-translational control to modify DKK1 protein abundance, and hence further regulates the precise level of Wnt signalling activity. Our finding provides a molecular mechanism linking the loss of the ubiquitin ligase NEDD4 in neural crest cells to

changes in DKK1 abundance. While we have focused on post-translational regulation of DKK1 protein, we also observe an increase in *Dkk1* mRNA in neural crest cells of *Wnt1-Cre; Nedd4^fl/fl* embryos. Indeed, many of the differentially expressed genes that were upregulated in *Wnt1-Cre; Nedd4^fl/fl* embryos are localised to the cardiac neural crest. This likely points to cell-autonomous roles for *Nedd4* loss-of-function in cardiac neural crest cells, which may also underpin cardiac defects. This would be a subject of future studies in heart development.

The clinical significance of our findings is highlighted by the outcome of the functional study of a deleterious homozygous missense variant of *NEDD4* in an individual with Tetralogy of Fallot. This NEDD4 variant results in a non-synonymous amino acid change in the protein domain responsible for ubiquitin ligase activity, and we show that this variant has a greatly reduced ability to ubiquitinate DKK1, as well as exhibiting heart defects in a CRISPR-engineered in vivo model. While the heart defects exhibited in *Nedd4^K766R/K766R* mice do not fully phenocopy the defects observed in *Wnt1-Cre; Nedd4^fl/fl* mice, this is likely due to functional differences between hypomorphic NEDD4 K766R protein expression in the whole embryo versus complete removal of NEDD4 protein in neural crest cells of *Wnt1-Cre; Nedd4^fl/fl* mice. The clinical Tetralogy of Fallot phenotype in the NEDD4 p.K766R individual overlaps with the heart defect phenotype of *Wnt1-Cre; Nedd4^fl/fl* mice, implicating an association of neural crest-derived NEDD4 and congenital heart disease. Mis-regulation of DKK1 activity may represent a new pathogenic mechanism underpinning congenital heart disease. Neural crest cells have never been considered as paracrine modifiers of Wnt signalling activity. We here define a new developmental paradigm in heart morphogenesis, and implicate neural crest cell modulation of Wnt signalling as a key mechanistic attribute controlling heart development.

## Methods
### Mice
All experiments were carried out in accordance with the ethical guidelines of the University of South Australia Animal Ethics Committee. Mice are housed with a 12 h dark/light cycle, with temperature maintained between 19-23 °C and 40-70% relative humidity. Mice used for timed matings were between 6-12 weeks of age. All mouse lines were maintained on a mixed C57BL/6 background. *Nedd4^-/-* mice and *Nedd4^fl/fl* have been described previously[49,50]. To remove *Nedd4* in specific tissues, we crossed *Nedd4^fl/fl* females to *Wnt1-Cre;Nedd4^fl/+*[51],

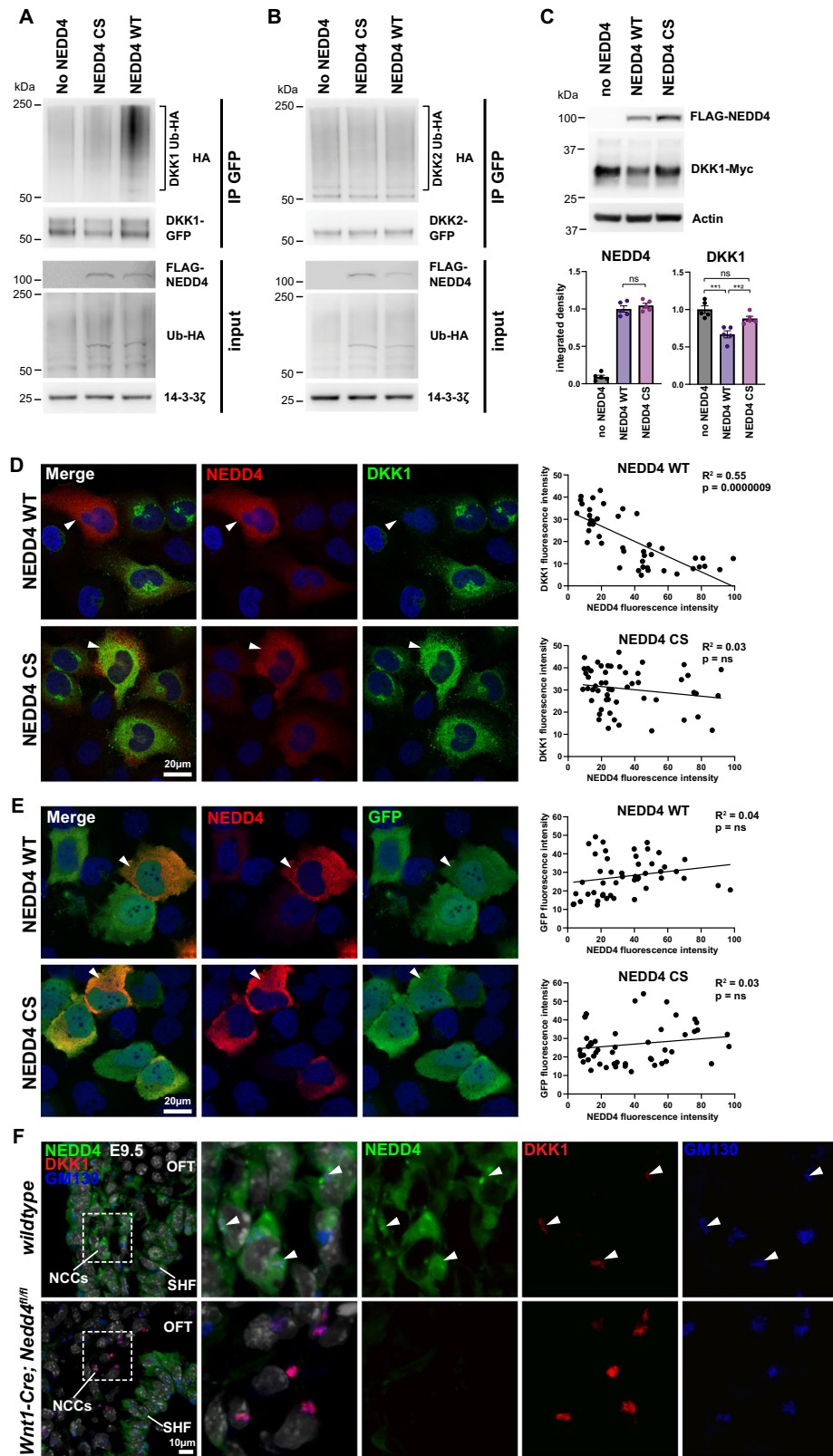

*Mef2cAHF-Cre;Nedd4*[fl/+52] or *Tie2-Cre;Nedd4*[fl/+53] males. *Wnt1-Cre; Nedd4*[fl/fl] mice were intercrossed with *Dkk1*[+/−47] mice (*Wnt1-Cre; Nedd4*[fl/+] males crossed to *Nedd4fl/fl; Dkk1*[+/−] females) to generate *Wnt1-Cre; Nedd4*[fl/fl]; *Dkk1*[+/−] embryos (as well as littermate controls) for assessing the rescue effect of DKK1 dosage compensation. The following reporter lines were also intercrossed with *Wnt1-Cre;Nedd4*[fl/+] mice; *Mef2cAHF-LacZ*[54] and *Z/EG*[55]. *Nedd4*[K766R] mice were generated using CRISPR targeting. As

embryos that contained the variant on both chromosomes (homozygotes) or in combination with a frameshifting indel (compound heterozygotes) were likely to have compromised cardiac function, we used a dual template competition strategy. SaCas9 KKH mRNA was microinjected with two single-stranded DNA oligonucleotide donor templates. The first oligonucleotide contained silent mutations at residues K766, I765 & L764 and retained WT function when integrated

**Fig. 6 | NEDD4 ubiquitinates DKK1 to regulate protein levels. A** Western blot of an in vitro ubiquitination assay. HEK293T cells were transfected with DKK1-GFP, Ub-HA, and either no NEDD4, FLAG-NEDD4 CS (cysteine mutant) or FLAG-NEDD4 WT (wild-type). Immunoprecipitation for GFP demonstrates HA positivity with NEDD4 WT, indicating ubiquitination of DKK1, but minimal ubiquitination with NEDD4 CS or no NEDD4. Representative image from $n = 3$ independent experiments. **B** Ubiquitination assay as in (**A**), but with DKK2-GFP construct. Minimal ubiquitination is observed in all conditions. Representative image from $n = 3$ independent experiments. **C** Western blot of HEK293T cells transfected with DKK1-Myc and FLAG-NEDD4 as indicated. Representative image from $n = 5$ biological experimental replicates. Quantitation of integrated density normalised to Actin shows that steady-state DKK1 protein levels are reduced with NEDD4 WT compared to either no NEDD4 or NEDD4 CS. ns not significant; $**1$ $P = 0.0012$; $**2$ $P = 0.0061$ (unpaired $t$ test, two-tailed). **D** HeLa cells transfected with DKK1-GFP and FLAG-NEDD4 WT or CS, immunostained for FLAG and DKK1. Representative images from $n = 3$ independent experiments. Mean fluorescence intensity of DKK1 and NEDD4 quantified for individual cells and plotted graphically shows a negative linear correlation between DKK1 and NEDD4 WT levels, but not NEDD4 CS (simple linear regression). For transfection control for DKK1 construct, see Supplementary Fig. 16. **E** HeLa cells transfected with GFP and FLAG-NEDD4 WT or CS, immunostained for FLAG and GFP. Representative images from $n = 3$ independent experiments. Quantification of mean fluorescence intensity reveals no correlation between NEDD4 and GFP levels (Simple linear regression). **F** Sagittal sections through the second heart field (SHF) region of E9.5 wild-type and *Wnt1-Cre; Nedd4*$^{fl/fl}$ embryos immunostained for NEDD4, DKK1, and GM130. Representative images from $n = 5$ independent embryos per genotype. Arrowheads indicate areas of co-localisation of NEDD4 with DKK1, in the neural crest cells (NCCs), which overlap with areas of the Golgi marker GM130. In *Wnt1-Cre; Nedd4*$^{fl/fl}$ embryos, where NEDD4 is absent, DKK1 expression is increased. Source data are provided as a Source Data file.

(sequence: TTCCGACGCGTTTCCTTTCCTTCTAGGGATTTTTTGAACT-GATACCACAGGATttgATtAAaATATTTGATGAAAATGAGCTA-GAGGTAAGAACTATTTCTGCATGTGCTTGGGAATGTAG). The second oligonucleotide encoded the K766R variant and a silent CTC(L764) > TTG(L764) mutation to abolish the PAM site and prevent re-cutting of the targeted DNA by the nuclease (sequence: TTCCGACGCGTTTCC TTTCCTTCTAGGGATTTTTTGAACTGATACCACAGGATTTGATCAGGA TATTTGATGAAAATGAGCTAGAGGTAAGAACTATTTCTGCATGTGC TTGGGAATGTAG). The K766R oligonucleotide also introduced a BclI recognition site for subsequent genotyping. Both oligonucleotides had 60 bp homology arms on either side of the targeted mutations.

To obtain mouse embryos of defined gestational ages, mice were mated in the evening, and the morning of vaginal plug formation was counted as E0.5. In all experiments, stage-matched embryo littermates were selected for comparison using stage-appropriate parameters such as somite number, crown-rump length, and forelimb/digit maturity. For embryonic analyses, we did not genotype for sex.

### Drug treatments of pregnant dams
Pregnant dams were intraperitoneally injected with 16.6 mg/kg CHIR99021 (Selleck Chem) or 10 mg/kg iCRT3 (Selleck Chem) at the indicated time points in 15% DMSO, 30% PEG 300, PBS vehicle. Embryos were harvested at E13.5 or E9.5, and processed for immunostaining.

### Histology and immunostaining
Embryos were fixed in 4% paraformaldehyde in PBS. Cryosections were stained with Hematoxylin and Eosin to classify heart defects at E15.5-E17.5. For immunolabelling, cryosections, whole embryos, or fixed cells were blocked in 10% DAKO block, 0.2% BSA, 0.2% Triton X-100 in PBS, and stained with the indicated primary antibodies. Antibodies used were rabbit anti-NEDD4 (Abcam 14592) 1:300; rabbit anti-NEDD4 (purified serum, gift from S. Kumar) 1:300; mouse anti-alpha smooth muscle actin (Sigma A2547) 1:2000; rat anti-CD31 (Biolegend 102502) 1:150; goat anti-SOX10 (R&D Systems AF2864) 1:200; chicken anti-GFP (Abcam ab13970) 1:1000; mouse anti-Isl1 (DSHB 39.4D5 or 40.3A4) 1:50; mouse anti-MF20 (DSHB) 1:100; goat anti-DKK1 (R&D Systems AF1765) 1:200; mouse anti-AP2a (DSHB 3B5) 1:20; rabbit anti-beta-catenin non-pS45 (Cell Signaling Technology 19807) 1:200; mouse anti-beta-catenin ABC (8E7) (Millipore 05-665) 1:100; mouse anti-beta-catenin pY489 (DSHB) 1:50; mouse anti-FLAG (Sigma F3165) 1:1000; mouse anti-GM130 (BD 610822) 1:100; rabbit anti-phospho-Histone H3 (Millipore 06-570) 1:500; rabbit anti-cleaved-Caspase-3 (Cell Signaling Technology 9661) 1:500; rabbit anti-phospho-SMAD1/5/9 (Cell Signaling Technology 13820) 1:200; rabbit anti-phospho-ERK1/2 (Cell Signaling Technology 4370) 1:100; goat anti-Scribble (Santa Cruz Biotechnology sc-11048) 1:50; rabbit anti-Laminin (Sigma L9393) 1:1000; rabbit anti-Fibronectin (DakoCytomation A0245) 1:1000; mouse anti-N-cadherin (Cell Signaling Technology 14215) 1:100;

AlexaFluor 647 conjugated Phalloidin (Invitrogen) 1:200. EdU staining was performed following manufacturers recommendations (Invitrogen Click-iT EdU AlexaFluor 555 Cell Proliferation Kit). Slides were mounted in Fluoro-mount G with DAPI (ProSciTech). Histology and whole embryo/heart images were captured using Olympus DP2-AOU1.1 software or Improvision OpenLab 5 software. Confocal images were acquired with a Zeiss LSM 800 microscope and ZEN 2.6 (Blue edition) software.

### Fluorescence intensity and image quantitation
Mean fluorescence intensity (MFI) was measured using ZEN 3.4 (Blue edition) software (Zeiss) and represents the total fluorescence intensity of the region of interest, normalised to the area of the region of interest. For MF20 measurements in the second heart field at E9.5, a line of 100 μm was drawn downwards perpendicular to the outflow tract, and the single layer of cells of the second heart field (pericardial wall) was traced around only to the end of the 100 μm line. Hence, an equivalent area of MF20 fluorescent intensity was measured across different embryo samples. A minimum of 2 and up to 4 sagittal sections per embryo (at the medial position relative to the left and right side of the second heart field) were used to calculate MFI as described above, and averaged. Individual points on graphs represent the mean value calculated from 1 embryo. A minimum of 5 and up to 12 embryos per genotype were used for MF20 quantitation. In addition, MF20 MFI in the second heart field was normalised to MFI measurements taken of the adjacent unaffected atria within the same acquired microscopy image, and presented in the Supplementary Figs. MFI was measured from a 100 μm length of MF20+ve atria tissue, with the normalised value representing MFI in the second heart field/MFI in atria.

For MF20 MFI measurements of SHF explants, the entire DAPI+ve region was selected, and MFI was measured in this region of interest. Individual values represent MFI for independent explants (i.e., biological replicates) of the indicated genotype.

For DKK1 MFI measurements, a minimum of three and up to six sagittal sections per embryo were measured and averaged. An area encompassing the cardiac neural crest cells was drawn using the boundaries of the basal sides of the pharyngeal endoderm and dorsal pericardial wall for width, and ~100 μm for height (area ~2000–3000 μm²). Given DKK1 is only expressed in cardiac neural crest cells in the pericardiac region, it was not possible to assess any other DKK1+ve areas of immunostaining for the purposes of normalisation to an unaffected structure.

For β-catenin immunostaining, high-magnification images were assessed, and co-localisation of β-catenin immunostaining with DAPI fluorescence was counted as nuclear β-catenin + ve. A 100 μm length of second heart field nuclei were counted, which ~20–25 nuclei per section. β-catenin + ve nuclei were represented as a percentage of total nuclei counted. A minimum of 3 and up to 8 sagittal sections were quantified per embryo and averaged. Individual points on graphs

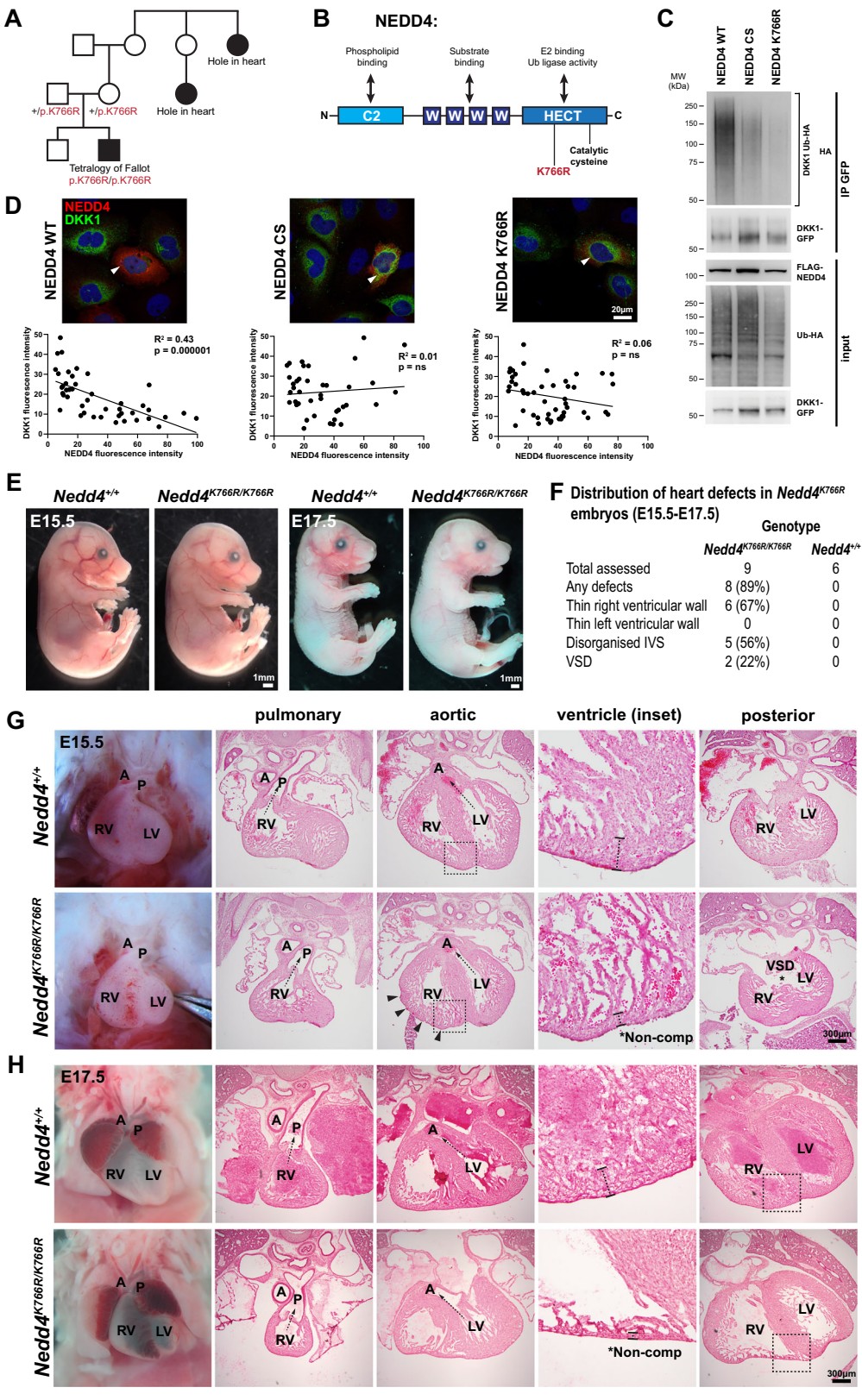

Nature Communications | (2026)17:1751                                                                                      13

represent the mean value calculated from 1 embryo. A minimum of 4 and up to 9 embryos per genotype were used for β-catenin quantitation.

For EdU quantitation in SHF after 1 h pulse, a 100 μm length of second heart field nuclei were counted, which ~20–25 nuclei per section. EdU +ve nuclei were represented as a percentage of total nuclei counted. Five sagittal sections were quantified per embryo and averaged.

Individual points on graphs represent the mean value calculated from 1 embryo, with 6 embryos assessed per genotype. For EdU migration front distance at E10.5 after 24 h chase, the EdU +ve migration front in the OFT was measured as the distance (in μm) of the last EdU +ve cell in the MF20+ve OFT from the pericardial wall. Three to four sagittal sections were quantified per embryo and averaged. Individual points on graphs represent the mean value calculated from 1 embryo. A minimum of 5

**Fig. 7 | A *NEDD4* variant identified in an individual with congenital heart disease has impaired ubiquitination of DKK1. A** Pedigree of family with a *NEDD4* variant. Genomic sequencing of the trio (parents and proband) revealed a homozygous *NEDD4* missense variant c.2297 A > G (p.K766R) in the proband with Tetralogy of Fallot. Hole in the heart was the term used by the family to describe distant relatives, with no clinical data available. **B** Protein domain structure of NEDD4, indicating the position of the K766R amino acid substitution. **C** Western blot of in vitro ubiquitination assay. Representative image from *n* = 3 independent experiments. HEK293T cells were co-transfected with DKK1-GFP, Ub-HA, and either NEDD4 WT, FLAG-NEDD4 CS (cysteine mutant), or FLAG-NEDD4 K766R. Immunoprecipitation for GFP demonstrates HA positivity with NEDD4 WT, indicating ubiquitination of DKK1, but minimal ubiquitination with NEDD4 CS or NEDD4 K766R. **D** HeLa cells transfected with DKK1-GFP and FLAG-NEDD4 WT, CS, or K766R, immunostained for FLAG and DKK1. Representative images from *n* = 3 independent experiments. Mean

fluorescence intensity of DKK1 and NEDD4 was quantified for individual cells and plotted graphically. Simple linear regression analysis reveals a negative correlation between DKK1 and NEDD4 WT levels, but not for NEDD4 CS or K766R. For transfection control for DKK1 construct, see Supplementary Fig. 16. ns not significant. **E** Representative image of control and *Nedd4*^K766R/K766R E15.5 and E17.5 embryos, from six independent litters. **F** Frequency of heart defects in E15.5-E17.5 *Nedd4*^K766R/K766R embryos, with defects noted in the right ventricle and interventricular septum. **G** Histological sections of control and *Nedd4*^K766R/K766R E15.5 embryos showing rugged surface of right ventricular wall (arrowheads), right ventricular non-compaction, and ventricular septal defect (VSD). Representative images from *n* = 3 *Nedd4*^+/+ and *n* = 4 *Nedd4*^K766R/K766R embryos. **H** Histological sections of control and *Nedd4*^K766R/K766R E17.5 embryos showing right ventricular non-compaction. Representative images from *n* = 3 *Nedd4*^+/+ and *n* = 5 *Nedd4*^K766R/K766R embryos. Source data are provided as a Source Data file.

and up to 6 embryos per genotype were used for quantitation. EdU +ve migration front distance was also normalised to atrio-ventricular valve primordium thickness (an unaffected structure present in the same microscopy images) as a means to account for any variation in embryo size, and is presented in the Supplementary Figs.

For HeLa cell assays, the entire cell outline was traced to define the region of interest, and MFI was measured for NEDD4 and DKK1 or GFP-only within this region of interest. Approximately eight images were acquired at random per transfection condition, from *n* = 3 experiments. Data points on graphs represent individual cells, with a minimum of 40 cells measured per transfection condition in total.

Semi-quantitative optical density measurements of whole-mount in situ hybridisation were performed in ImageJ. Images were subjected to colour deconvolution to generate 8-bit grayscale images of BCIP/NBT substrate staining. Mean grey values were calculated from a region of interest corresponding to the second heart field/dorsal pericardial wall. Optical density was calculated using the formula $OD = \log_{10}(255/\text{mean grey value})$. A minimum of three and up to seven whole-mount embryos per genotype were assessed.

### In situ hybridisation
Whole embryos or cryosections were hybridised with digoxigenin-labelled antisense probes for the following genes *Dkk1, Dlx2, Prrx1, Msx1, Msx2, Col1a1, Axin2, SP5, Pitx2c, Fam49a, Mical2, Tbx1, Tbx20, Isl1, HoxB1, Fgf8, Bmp4*. Signal was detected using anti-DIG-AP conjugated Fab fragments (Roche), and staining with NBT/BCIP (Roche). Whole embryos were imaged with Olympus CZX10, and tissue sections were imaged with Olympus IX73.

### β-galactosidase staining
Fixed embryos were incubated in staining solution: 19 mM sodium dihydrogen phosphate, 81 mM Disodium hydrogen phosphate, 2 mM $MgCl_2$, 5 mM EGTA, 0.01% sodium deoxycholate, 0.02% NP-40, 5 mM potassium ferricyanide, 5 mM potassium ferrocyanide, and 1 mg/ml X-gal substrate, at 37 °C until blue staining was sufficient.

### SHF explant
The second heart field region was dissected from E9.5 embryos by removing the head above pharyngeal arch 2, cutting off the outflow tract and heart, cutting down the lateral sides of the pharynx to create a tissue flap, and then separating this second heart field tissue flap at the posterior end from the rest of the embryo. This was transferred to tissue culture dishes coated with 50 μg/ml Collagen and 50 μg/ml Fibronectin, and cultured for 5 days in 5% FCS + DMEM. Explants were fixed in 4% PFA.

### SHF deployment assay
In methods similar to refs. 17,18, pregnant dams were intraperitoneally injected with 100 mg/kg EdU (5-ethynyl-2′-deoxyuridine) at E9.5 to pulse-label proliferating cells. Embryos were harvested after 1 h to

assess the extent of EdU labelling. To chase EdU-labelled proliferating cells, pregnant dams were injected with 500 mg/kg of thymidine 1 h after the EdU pulsing period to compete for remaining EdU label, and were harvested 24 h later at E10.5. Embryos or tissue sections were stained with Click-iT EdU Kit AlexaFluor-555 conjugated (Life Technologies) following staining with primary antibodies.

### Laser capture and RNA preparation
Tissue preparation and laser capture were based on ref. 56. Briefly, mouse embryos were fresh-frozen in OCT, cryosectioned, and collected on PEN membrane slides. Slides were stained with 1% Cresyl Violet to visualise tissue sections, air-dried, and then laser dissection was performed promptly (Leica LMD) to specifically capture the central anterior second heart field region (with associated neural crest and pharyngeal endoderm tissue). RNA was prepared from tissue by lysing in 4 M guanidine isothiocyanate solution (Invitrogen) and precipitating with sodium acetate and ethanol. After 75% ethanol wash, the RNA pellet was resuspended in 12 μl $H_2O$, with 1 μl reserved for Bioanalyser quality control of RNA, and 9.5 μl used for subsequent cDNA synthesis.

### RNA sequencing and bioinformatics analyses
Isolated RNA samples were submitted to the ACRF Genomics Facility for library preparation and sequencing. Extracted RNA was quantified using the Bioanalyser Pico chip, and 9.5 μl (containing 5–17 ng of RNA) was input into SMART-Seq v4 Ultra Low Input RNA Kit for Sequencing (Takara Bio) for generation and amplification of double-stranded full-length cDNA (using nine cycles of LD-PCR). Prior to library preparation, cDNA was sheared to 200–500 bp fragments using the Covaris system. NGS libraries were prepared using the ThruPLEX DNA-seq Kit (Takara Bio) together with the DNA Unique Dual Index Kit (Takara Bio), using 11 cycles of PCR for library amplification. Indexed libraries from three biological replicates for each *wildtype* and *Wnt1-Cre; Nedd4*^fl/fl samples were pooled equally for sequencing using 2 × 75bp reads on an Illumina NextSeq High Output kit. Raw data, averaging 47 million reads per sample, were analysed and quality checked using the FastQC program (http://www.bioinformatics.babraham.ac.uk/projects/fastqc). Reads were mapped against the mouse reference genome (mm10) using the STAR spliced alignment algorithm[57] (version 2.6.1 d with default parameters and --chimSegmentMin 20, --quantMode GeneCounts), returning an average unique alignment rate of 81%. The resulting mapped reads were found to possess a high rate of duplicates (≈ 64%). These were marked and subsequently removed using Picard Tools (version 2.16.0)[58] leaving, on average 18 million deduplicated reads per sample. Differential expression analysis was evaluated from TMM normalised gene counts using R (version 3.2.3), edgeR (version 3.3)[59] and Degust[60] following protocols as described[61]. Alignments were visualised and interrogated using the Integrative Genomics Viewer v2.3.80[62]. Graphical representations of differentially expressed genes were generated using Glimma[63], Degust[60], and MaGIC Volcano Plot Tool[64]. Gene ontology analysis was performed using DAVID[65,66].

## Ubiquitination assays

For cell-based ubiquitination assays, HEK293T cells originally obtained from ATCC (CRL-3216) were grown in DMEM supplemented with 10% foetal calf serum. Plasmid transfections were carried out with Lipofectamine 2000 (Thermo Fisher). HEK293T cells were co-transfected with an equal amount of pcDNA3-FLAG-tagged NEDD4 WT, NEDD4 CS, or NEDD4 K766R, pCDNA3-HA-tagged ubiquitin (gift from Sharad Kumar), and GFP-tagged DKK1 or DKK2. The DKK1-GFP and DKK2-GFP constructs were generated by cloning a PCR fragment of the open reading frames from pCS2-DKK1-FLAG (Addgene plasmid #16690) or pCS2-DKK2-FLAG (Addgene plasmid #15495) in frame to EGFPN3 using EcoRI and BamHI restriction sites. Cells were grown for 24–48 h at 37 °C with 5% $CO_2$ in a humidified atmosphere. Transfected cells were treated with 10 µM MG132 and 100 µM of Chloroquine for 3 h prior to protein extraction.

## Immunoprecipitation and western blot

Cells were lysed in NP-40 buffer (137 mM NaCl, 10 mM Tris pH 7.4 (108319 Merck), 10% glycerol, 1% NP-40, 2 mM sodium fluoride, and 2 mM sodium vanadate) containing Complete Protease Inhibitors. For purification of GFP-fusion proteins, GFP-Trap beads were used following the manufacturer's protocols (gta-10 Chromotek). Protein samples were separated by SDS-PAGE and transferred to a PVDF membrane (88585 Thermo Fisher). Membranes were blocked in 5% skim milk powder in tris-buffered saline (TBS, 50 mM Tris pH 7.5, 150 mM NaCl) with 0.1% Tween 20 (P1379 Merck). The following primary antibodies were used: mouse anti-HA (6E2, Cell Signaling Technologies 2367) 1:1000; mouse anti-Myc (9B11, Cell Signaling Technologies 2276) 1:1000; rabbit anti-GFP (ClonTech 632592) 1:1000; Goat anti-DKK1 (R&D Systems AF1765) 1:500; mouse anti-beta-Actin (Sigma A5441) 1:1000; rabbit anti-14-3-3zeta C-16 (sc1019 Santa Cruz) 1:1000. Western blot images were captured using BioRad Image Lab Touch Software or FUJIFILM ImageQuant LAS 4000 software. Densitometry of all western blot data was performed with ImageJ.

## HeLa protein abundance correlation assays

HeLa cells (kindly provided by the laboratory of Prof. Stuart Pitson (Centre for Cancer Biology, Adelaide) were grown in DMEM supplemented with 10% foetal calf serum. Plasmid transfections were carried out with X-tremeGENE HP (Roche) in eight-well glass-bottom imaging slides (Eppendorf). Cells were co-transfected with an equal amount of FLAG-tagged NEDD4 WT, NEDD4 CS, or NEDD4 K766R, and DKK1-GFP. Cells were grown for 24–48 h at 37 °C with 5% $CO_2$ in a humidified atmosphere. Cells were fixed with 4% paraformaldehyde in PBS for 20 min, and immunostained following the protocol described above. Mean fluorescent intensity for each cell was calculated using ZEN 3.4 software (Zeiss). Approximately 15 cells were quantified for each condition per experiment, for $n = 3$ individual experiments. Graphs represent combined data for all experiments. Simple linear regression analysis was performed in GraphPad Prism.

## Human patient recruitment

Ethics approval was obtained from the Sydney Children's Hospital Network Human Research Ethics Committee (approval number HREC/16/SCHN/73). The family was recruited through the Kids Heart Research DNA Bank at the Children's Hospital at Westmead, Sydney, Australia. Informed, written consent was obtained from the family. Heart defects in the proband were confirmed by echocardiography.

## Patient sequencing and genome data analysis

Genomic DNA was extracted, as described previously[67]. DNA sample libraries were prepared using the Illumina TruSeq Nano DNA HT Library Prep Kit and genome sequenced (Illumina HiSeq X Ten, 150 base paired-end reads) at Genome.One, Garvan Institute of Medical Research, Sydney, Australia. Calling and annotation of sequencing data

were performed as per ref. 29. In short, sequence reads were mapped to the human reference genome hg38 using Burrows-Wheeler Aligner (BWA-mem v0.7.12)[68]. Single-nucleotide variants (SNVs) and small insertions or deletions were called using Platypus v0.8.1[69]. The variant call files (VCFs) were annotated using ANNOVAR v 2016Feb01[70]. Variants were prioritised as described in ref. 29 by screening for a list of curated CHD genes, followed by an unbiased comprehensive analysis for all genes. Variant frequencies were in reference to the gnomAD database (v2.1.1)[71]. A variant of interest identified in the family, NEDD4:c.2297 A > G:p.K766R (GenBank:NM_006154.4), had an allele frequency of 0.004438.

## Statistics

In all graphical representations, individual values represent independent biological replicates. Unless otherwise specified, $P$ values were calculated using unpaired two-tailed $t$ test. For grouped analysis, significance was assessed using 1 or two-way ANOVA multiple comparisons. All graphs and statistical analyses were performed using GraphPad Prism 10.

## Reporting summary

Further information on research design is available in the Nature Portfolio Reporting Summary linked to this article.

## Data availability

Laser capture mRNA sequencing data is available publicly from NCBI Gene Expression Omnibus under accession number GSE309818. All other data supporting the findings of this study are available within the article or Supplementary Information files or from the corresponding authors upon request. Source data are provided with this paper.

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

## Acknowledgements

This research was supported by funds from the National Health and Medical Research Council (NHMRC): Project Grant (ID1144004) (Q.S. and S.W.); Ideas Grant (ID2030340) (S.W. and Q.S.); Principal Research Fellowship (ID1135886) (S.L.D.); Leadership Level 3 Fellowship (ID2007896) (S.L.D.); Project Grant (ID1162878) (S.L.D., D.W., and E.G.); Synergy Grant (ID1181325) (S.L.D., D.W., and E.G.). Channel 7 Children's Research Foundation (S.W.). Royal Adelaide Hospital Mary Overton Postdoctoral Fellowship (S.W.). National Heart Foundation Future Leader Fellowship (Q.S.). NSW Health Cardiovascular Research Capacity Program Senior Researcher Grant (S.L.D.). We thank Paul Riley for the critical reading of the manuscript.

## Author contributions

S.W. and Q.S. designed the study and experiments. S.W. and Q.S. wrote the manuscript. S.W., I.L., C.M., G.S., D.D., J.H., T.Z., M.T., and Q.S. performed all wet laboratory experiments. G.M.B. and D.W. recruited the patient and family with congenital heart disease. D.A., E.G., and S.L.D. performed and analysed whole-genome sequencing data and identified the NEDD4 variant. W.P. prepared libraries for laser capture mRNA sequencing. J.T. performed bioinformatic analysis of laser capture mRNAseq data. M.W., S.P., and P.Th. generated NEDD4K766R mice. PPLT provided *Dkk1*⁻/⁻ mice. NH provided *Tie2-Cre; Nedd4*^fl/fl^ mice. All authors intellectually contributed to and approved the final manuscript.

## Competing interests

The authors declare no competing interests.

## Additional information

## Congenital Heart Disease Synergy Group

Sally L. Dunwoodie ⑩ ³,⁸, David Winlaw ⑩ ⁹,¹⁰, Eleni Giannoulatou ⑩ ³,⁸, Edwin Kirk¹⁵,¹⁶,¹⁷, Gavin Chapman³,¹⁵, Natasha Nassar¹⁸, Gillian M. Blue ⑩ ¹¹,¹², Gary Sholler¹¹,¹² & Samantha Lain¹⁸

¹⁵School of Clinical Medicine, University of New South Wales, Kensington, NSW, Australia. ¹⁶Centre for Clinical Genetics, Sydney Children's Hospital, Randwick, NSW, Australia. ¹⁷Randwick Genomics Laboratory, NSW Health Pathology, Randwick, NSW, Australia. ¹⁸Children's Hospital Westmead Clinical School, Faculty of Medicine and Health, The University of Sydney, Westmead, NSW, Australia.

