## [Peer Review file · Nature Communications]

Neural crest cell-derived DKK1 and NEDD4 modulate Wnt signalling in the second heart field to orchestrate outflow tract development

Corresponding Author: Dr Sophie Wiszniak

Version 0:

Reviewer comments:

Reviewer #1

(Remarks to the Author)

In this manuscript, Wiszniak et al. describe a striking phenotype resulting from the ablation of Nedd4 specifically in neural crest cells. The authors report that this genetic alteration leads to premature differentiation of second heart field (SHF) progenitors, resulting in outflow tract (OFT) defects, including shortening and mispositioning. Mechanistically, these defects are attributed to upregulated expression of DKK1 in neural crest cells, which under normal conditions express this gene at lower levels. This, in turn, leads to reduced Wnt signaling within the SHF and consequently, premature differentiation of its progenitors.

While the research is well conducted and supported by a strong genetic approach, a more direct elucidation of neural crest cell function during normal OFT development would require in vivo evidence that NEDD4-mediated ubiquitination and degradation of DKK1 occurs specifically within neural crest cells.

I have a few additional comments:

1. Could the authors provide detailed data on Nedd4 expression? Is it expressed in branchial arches, the heart, and neural crest cells?
2. In Fig 1, Could the authors comment on differences in distributions of OFT defects observed in Nedd4^{-/-} vs Wnt1-Cre; Nedd4^{fl/fl} embryos?
3. Fig 3C, why is there no apparent loss of Isl1+ SHF tissue, given the precocious differentiation into myocardium?
4. Why is enhanced MF20 expression interpreted as a marker of premature SHF differentiation? Please clarify for the readers.
5. In Figure 4H, β -catenin appears to localise primarily at cell-cell junctions rather than in the nucleus. In the mutants, the reduction is most apparent at the junctional level. The reviewer does not observe a reduction in nuclear β -catenin. Could the authors clarify this, possibly by providing higher magnification images? The evidence for reduced Wnt signaling in the SHF, based only on β -catenin staining, is not fully convincing. Could the authors validate this using additional Wnt reporters such as TCF/Lef:H2B-GFP, Axin2-lacZ, etc..?
6. Could the authors discuss why enhancing Wnt signaling with CHIR99021, a well-known Wnt pathway activator, failed to rescue the OFT defects in Wnt1-Cre; Nedd4^{fl/fl} embryos? Could the author analyse Wnt signalling activity under CHIR99021 and iCRT3 conditions?
7. In Figure 4G, there appears to be no upregulation of MF20 in Wnt1-Cre; Nedd4^{fl/fl} embryos. Could the authors clarify this inconsistency?
8. In Figure 4G, the concept of an "EdU migration front" is unclear. How is this defined, and how does it relate to the extent of SHF incorporation into the myocardium?

9. Lastly, could the authors discuss the phenotypic differences between Nedd4K766R embryos and Nedd4^{fl/fl} embryos, and explain why the K766R mutation, which replicates a human variant, does not phenocopy neural crest-specific loss of Nedd4?

Reviewer #2

(Remarks to the Author)

In the manuscript "Neural crest cell-derived DKK1 modulates Wnt signalling in the second heart field to orchestrate cardiac outflow tract development," Wiszniak et al examine the role of NEDD4 expressed in neural crest cells during cardiac development. The authors demonstrate that the deletion of NEDD4 results in defects in the development of the cardiac outflow tract and the positioning of the great arteries relative to the right and left ventricles. The authors examine the evidence that this effect is due to increased levels of Dkk1 in their mutants. While the idea is very interesting, the data does not support the authors' conclusions. In many experiments, the experimental setup is not appropriate for what the authors intended to study.

Major Concerns:

1) A major concern is that for multiple types of experiments, normalization is not provided.

a) For example, the length of the outflow tract is not normalized to an unaffected embryonic structure (e.g., otic vesicle) to control for embryo size in Figure 2A. In addition, it is not clear how authors consistently found a boundary between the proximal OFT and the right ventricle. This needs to be described. Alternatively, the RV needs to be included in the measurements because it is difficult to find such a boundary, and customarily, the RV is included in the measurements of OFT length at E10.5.

b) Another example is the measurement of valve diameter in Figure 5G. This measurement also needs to be normalized to an unaffected embryonic structure to control for embryo size. These normalizations are essential since no single embryo develops at the same rate, and differences in embryo stages and sizes are common even within the same litter.

c) It is not proper to report fluorescence intensity as a measurement without normalization, since fluorescence intensity will vary based on many factors beyond the experimenter's control, such as fluctuations in laser power, differences in cover slip thickness, and the exact plane of focus. Check the Bioimaging guide for best practices (https://www.bioimagingguide.org/03_Image_analysis/Intensity.html)

2) Lines through the ventricles seem to be drawn in random orientation in the mutants in Figure 5. While in controls, the line is drawn through the lumens of LV and RV, in the mutants, the line does not appear to go through the lumen of LV. Other section levels or coronal sections may be more appropriate. Multiple levels like those shown in Fig. 2B may also work well to demonstrate altered alignment.

3) On line 14, page 6, the authors state that NC numbers are not affected in the mutants. Since this is very important for the rest of the study, the authors need to provide quantification to support this statement.

4) On line 19, page 6, the authors state that the expression of additional SHF markers were not affected in the mutants. However, Sup. Fig 2 shows that Hoxb1 and BMP4 are downregulated and Fgf8 is upregulated in the mutants. Please state how many times each experiment was conducted; how many controls and mutants were analyzed for each staining, and provide quantification.

5) For the experiment with the explant in Fig. 3E, FACS is more appropriate to determine the proportion of MF20+ cells and intensity. Western blot is also appropriate, but the measurements shown are not up to the accepted standards for quantification.

6) The EdU experiment in Fig. 3G is not appropriate to show cell migration. For this experiment, EdU pulse-chase needs to be implemented: after one hour of EdU, thymidine needs to be injected to compete EdU off. Otherwise, the experiment in 5G can be interpreted as a decrease in proliferation rather than migration. See doi: 10.1016/j.ydbio.2016.02.017 as an example of such an assay in vivo. Also see papers by Elaine Fuchs, as additional examples for how this is done.

7) Figure 4H. There is no nuclear β -catenin in controls or mutants. It is not clear what is being measured and how.

8) Figure 4K: Expression of Axin2 can be used as a readout for Wnt signaling. However, the authors point to Axin2 staining in the atrial chamber in a control embryo (arrowhead) and not the SHF in the dorsal pericardial wall contiguous with the OFT. The SHF appears not to express Axin2 in the control. In the mutant shown, the Axin2 staining did not seem to work because there is no staining in the atrial chamber, a tissue unaffected by the neural crest. This is an important experiment, and it needs to be done properly to determine whether Wnt signaling is indeed downregulated in the SHF in the mutants.

9) Figure 5E: MF20 IF intensity is reported without normalization and thus is unreliable. In addition to properly measuring and normalizing MF20 intensity, the authors should measure the length of SHF occupied by the ectopic MF20+ cells.

10) Figure 6: How do authors explain the mechanism by which NEDD4 regulates the ubiquitination of Dkk1? Dkk1 is a secreted protein (i.e., it is present inside the ER, Golgi, and secretory vesicles), while NEDD4 is a cytoplasmic protein that is always outside the ER, Golgi, etc. These two proteins are never in the same cellular compartment.

11) Figure 7G. The hearts look smaller in the mutants, suggesting that VSD and other "defects" may be due to delayed embryo development at E15.5 in the mutants. E18.5 is a more appropriate time point for the evaluation of structural heart defects.

a. Are Nedd4-K766R homozygous mutants viable?

Minor:

1) on line 14 page 4, the authors state that "neural crest cells form structural components of the outflow tract valves." This is incorrect. Neural crest cells die at late gestation and do not contribute to mature outflow tract valves. See papers from

Jonathan Epstein lab.

2) On line 19-21 page 4, the authors state "However, prior models in which neural crest cells have been surgically or genetically ablated⁴⁻⁷ have not been amenable to elucidate how neural crest cells interact with the second heart field to control its growth, differentiation and morphogenesis." This is inaccurate. The authors should mention the work from Margaret Kirby's lab showing the importance of NC in regulating BMP4-FGF8 cross talk between NCCs and SHF in preventing precocious differentiation of cardiac progenitors in the SHF.

Reviewer #3

(Remarks to the Author)

The MS by Wiszniak et al reports a novel regulatory mechanism whereby the ubiquitin ligase NEDD4 affects the protein levels of DKK1 in neural crest cells that modulates Wnt signaling thereby affecting the timing of second heart field differentiation and outflow tract morphogenesis. A combination of in vivo genetic studies in mice and in vitro examination of NEDD4 function provide convincing evidence for the major conclusions of the study. Notably, a DKK1 deficient mouse is provides evidence for a rescue of the NEDD4 deficiency in neural crest cells which is important support for the proposed mechanism. In addition, variants in NEDD4 were identified in individuals with congenital heart malformations and loss of function demonstrated experimentally.

This is a strong study with convincing data to support a new regulatory contribution of Wnt signaling in cardiac outflow tract development with human clinical implications.

Minor comments

1. The title does not mention NEDD4 which is a major focus of the study.
2. Were abnormalities in outflow tract endocardial cushions or semilunar valve primordia observed?
3. The Discussion is long and includes information on BMP and FGF signaling that seem peripheral to the current study.

Version 1:

Reviewer comments:

Reviewer #1

(Remarks to the Author)

I am satisfied with the authors' response to my previous comment. Overall, this is an insightful and compelling study with strong genetic evidence. For clarity, I would still recommend moving Supplementary Figure 4A and 4B into the main figures.

Reviewer #2

(Remarks to the Author)

The revisions of the manuscript entitled "Neural crest cell derived DKK1 and NEDD4 modulate Wnt signalling in the second heart field to orchestrate cardiac outflow tract development," has resulted in a really beautiful manuscript with high-quality data reporting novel and interesting findings that are also relevant to human congenital heart disease. The revised manuscript has addressed all previous critiques. The following are minor concerns remaining:

- 1) Reviewer Figure 3 - Axin2 in situ hybridization should be included in the manuscript.
- 2) Please mention which Wnt1-Cre strain was used? Wnt1-Cre1 or Wnt1-Cre2? Wnt1-Cre1 ectopically expresses large amounts of Wnt1 protein (Lewis AE, Vasudevan HN, O'Neill AK, Soriano P, & Bush JO (2013). The widely used Wnt1-Cre transgene causes developmental phenotypes by ectopic activation of Wnt signaling. *Dev Biol*, 379(2), 229–234. 10.1016/j.ydbio.2013.04.026).
- 3) In the methods, please add the genetic background of each parental strain used. In general, it would be useful to know the parental genotypes of the crosses that were used to generate various embryo genotypes. This information should be added to the methods as well.
Please add to the methods whether Wnt1-Cre; Nedd4fl/fl; Dkk1 +/- embryos used for quantifications in Fig. 5G are littermates of Wnt1-Cre; Nedd4fl/fl; Dkk1 +/- . If they were not littermates, please note whether or not differences in phenotypes between Wnt1-Cre; Nedd4fl/fl; Dkk1 +/- and Wnt1-Cre; Nedd4fl/fl; Dkk1 +/- may be due to potential differences in genetic background.
- 4) Please add a reference to your EdU pulse-chase method
- 5) Sup. Fig. 8: pErk1/2 in staining in Sup. Fig. 8 is too dim and is hard to see.

Response to Reviewers

Manuscript # NCOMMS-25-39806-T (Wiszniak et. al.)

We thank the reviewers for their time and expertise in reviewing our manuscript. Please find our detailed responses to the reviewers concerns and Reviewer Figures 1-3 below.

REVIEWER COMMENTS

Reviewer #1 (Remarks to the Author):

In this manuscript, Wiszniak et al. describe a striking phenotype resulting from the ablation of *Nedd4* specifically in neural crest cells. The authors report that this genetic alteration leads to premature differentiation of second heart field (SHF) progenitors, resulting in outflow tract (OFT) defects, including shortening and mispositioning. Mechanistically, these defects are attributed to upregulated expression of *DKK1* in neural crest cells, which under normal conditions express this gene at lower levels. This, in turn, leads to reduced Wnt signaling within the SHF and consequently, premature differentiation of its progenitors.

While the research is well conducted and supported by a strong genetic approach, a more direct elucidation of neural crest cell function during normal OFT development would require *in vivo* evidence that *NEDD4*-mediated ubiquitination and degradation of *DKK1* occurs specifically within neural crest cells.

Assays to investigate protein ubiquitination typically require overexpression of constructs in cell lines to yield very high levels of expression, combined with MG132/Chloroquine treatment to inhibit protein degradation, with 1-2mg of total cell lysate often used as input into immunoprecipitations for *in vitro* ubiquitination assays. While we demonstrate that *NEDD4* ubiquitinates and degrades *DKK1* in several *in vitro* assays (eg. overexpression experiments in HEK293 and HeLa cells, Figure 6A,D) demonstrating ubiquitination *in vivo* remains challenging. It is not possible to yield enough protein lysate from E9.5 neural crest cells *in vivo* to perform any such ubiquitin assay/western blot/TUBE assay. We have developed a protocol to generate neural crest cells from human iPSCs, with good success after a standardised 10 day differentiation protocol. scRNAseq of these iPSC-derived NCCs reveals broad expression of markers of NCCs including *TFAP2A* (*AP2 α*) and *SOX10* (Reviewer Fig. 1A). *DKK1* is only expressed in a subset of neural crest cells *in vivo* at E9.5 (specifically the cardiac neural crest cells (Figure 4D,E, Supp. Fig. 10, and Reviewer Fig. 1C E9.5 cluster 14), and indeed we find that a small proportion of iPSC-derived NCCs express *DKK1* (Reviewer Fig. 1A cluster 12) suggesting that these likely correspond to cardiac NCCs. Furthermore, these cells share expression profiles of *DKK1*+ve cardiac NCCs *in vivo* (eg. *SOX10* -ve, *HAND1*+ve, *HAND2*+ve, *GATA3*+ve). While we have attempted to perform some *in vitro* assays using these iPSC-derived NCCs, the proportion of cells expressing *DKK1* is too low to yield enough experimental material to proceed with any successful assays (eg. Reviewer Fig. 1B). We are continuing with development of our iPSC-derived NCC differentiation protocol, and hope in future to enrich for *DKK1*+ve cardiac neural crest

cell populations, for which there is no currently published protocol and is a missing tool in the cardiac neural crest cell research field.

Towards providing “*in vivo evidence that NEDD4-mediated ubiquitination and degradation of DKK1 occurs specifically within neural crest cells*”, we now present additional figures to clarify the cardiac neural crest cell-specific expression pattern of DKK1. In Supplementary Fig. 10 we show that DKK1 is only expressed in neural crest cells via co-localisation with the neural crest cell marker AP2 α ; hence if NEDD4 is regulating DKK1, this must be only occurring in neural crest cells. Furthermore, DKK1 is not expressed in all neural crest cells, but rather only becomes expressed once neural crest cells have reached the second heart field region in their migration trajectory (Supp. Fig. 10), suggesting specificity of DKK1 expression to the cardiac neural crest. We have now also performed immunostaining for NEDD4 and DKK1 in cardiac neural crest cells *in vivo* (Figure 6F), and demonstrate subcellular co-localisation, suggesting that it is feasible that NEDD4 could be interacting with and ubiquitinating DKK1 in neural crest cells *in vivo*. Interestingly, NEDD4 demonstrates a particularly heightened and punctate expression pattern, specifically in cardiac neural crest cells (Supp. Fig. 1E), suggesting NEDD4 may play unique functional roles in this cell type.

I have a few additional comments:

1. Could the authors provide detailed data on *Nedd4* expression? Is it expressed in branchial arches, the heart, and neural crest cells?

We have previously documented expression of *Nedd4* in the branchial arches and neural crest in our publication (Wiszniak et. al. 2013, Dev Biol). Additionally, we have now performed detailed immunostaining for NEDD4 at E9.5, which we present in Supplementary Figure 1. This demonstrates broad expression of NEDD4 throughout all tissues including neural crest cells, pharyngeal arches, heart and second heart field. We also show that expression of NEDD4 is specifically absent in neural crest cells of *Wnt1-Cre; Nedd4^{fl/fl}* embryos. Notably, this new data identifies heightened expression and altered localisation of NEDD4 in cardiac neural crest cells, which was previously unknown.

2. In Fig 1, Could the authors comment on differences in distributions of OFT defects observed in *Nedd4*^{-/-} vs *Wnt1-Cre; Nedd4^{fl/fl}* embryos?

While the distribution of OFT defects differs between *Nedd4*^{-/-} and *Wnt1-Cre; Nedd4^{fl/fl}* embryos, the broader class of OFT defect observed is the same, ie. artery-ventricle alignment defects. We have now modified a sentence in the manuscript to emphasise this more clearly (“*Importantly, the outflow tract defects identified in *Nedd4*^{-/-} and *Wnt1-Cre; Nedd4^{fl/fl}* embryos are of the same class of outflow tract defect, that being artery-ventricle alignment defects, that present on a variable spectrum of severity.*”). Heart defects often present on a scale of variable severity. In terms of artery-ventricle alignment defects, severity can range due to the degree of defective outflow tract rotation. A small reduction in OFT rotation may lead to overriding aorta (where the aorta is positioned between right and left ventricles, but pulmonary is normal), further reduction in OFT rotation may lead to DORV (where the aorta and pulmonary are both

aligned with the right ventricle), with greatly reduced OFT rotation leading to TGA (where the aorta and pulmonary arteries are completely switched ie. aorta aligns with right ventricle and pulmonary with left ventricle). PTA is the most severe defect, with no OFT rotation and no aorticopulmonary septum, leading to a single OFT vessel connecting to the heart, and straddling both ventricles. The distribution of OFT defects observed in *Nedd4^{-/-}* and *Wnt1-Cre; Nedd4^{fl/fl}* embryos is consistent with *Nedd4^{-/-}* embryos exhibiting defects at the very severe end of this spectrum, while *Wnt1-Cre; Nedd4^{fl/fl}* embryos exhibit defects at the moderate-severe end.

Nedd4^{-/-} embryos have severe growth retardation, and from E12.5 onwards are approximately half the size of their wildtype littermates (Cao et. al. 2010, Science Signaling 1:ra5, and Reviewer Fig. 2). We hypothesise that these severe systemic defects in whole embryo development likely impact on heart development, and as such leads to OFT rotation defects that are more pronounced. There may also be important roles for NEDD4 in other tissue lineages that contribute to heart development that were not tested in our conditional knockout analyses, such as the first heart field or pharyngeal mesoderm.

3. Fig 3C, why is there no apparent loss of Isl1+ SHF tissue, given the precocious differentiation into myocardium?

Isl1 is expressed in the SHF, but is also expressed in MF20+ve myocardium of the outflow tract (See wildtype examples in Figure 3B and 3C). Isl1 immunostaining only diminishes once myocardial cells have reached the right ventricle. Hence we do not expect to see any reduction in Isl1 in the SHF given the precocious MF20+ve differentiation in *Wnt1-Cre; Nedd4^{fl/fl}* embryos.

4. Why is enhanced MF20 expression interpreted as a marker of premature SHF differentiation? Please clarify for the readers.

We now add a sentence to our manuscript to clarify for the readers why enhanced MF20 expression is considered a marker of premature SHF differentiation, and reference other publications that have used MF20 immunostaining for this analysis.

“Myocardial differentiation is initiated as second heart field cardiac progenitors enter the developing outflow tract, and upregulate expression of various myocardial markers such as myosin heavy and light chains and other myogenic factors¹³. MF20, which recognises myosin heavy chain, is widely used to assess in vivo differentiation dynamics at the transition zone between the second heart field and outflow tract myocardium¹³⁻¹⁶.”

5. In Figure 4H, β -catenin appears to localise primarily at cell-cell junctions rather than in the nucleus. In the mutants, the reduction is most apparent at the junctional level. The reviewer does not observe a reduction in nuclear β -catenin. Could the authors clarify this, possibly by providing higher magnification images? The evidence for reduced Wnt signaling in the SHF, based only on β -catenin staining, is not fully convincing. Could the authors validate this using additional Wnt reporters such as TCF/Lef:H2B-GFP, Axin2-lacZ, etc..?

We now present higher magnification images of the β -catenin (non-phospho S45) immunostaining shown in Figure 4H in Supplementary Figure 11A. Furthermore, we have now highlighted examples of nuclear localised β -catenin (solid arrowheads) vs examples of no nuclear β -catenin (outlined arrowheads) to make our quantitation methods clearer for the reader.

In addition to the previous immunostaining shown for β -catenin non-phospho S45 (which is considered an active form of β -catenin), we have now performed additional immunostaining using another two antibodies that react to active forms of β -catenin. These include Active Beta Catenin (ABC) (clone 8E7), and β -catenin phospho Y489, and are presented and quantified in Supplementary Figure 11C-F. We have also examined the canonical Wnt signalling target gene *Pitx2* by immunostaining for Pitx2 protein, which we present and quantify in Supplementary Figure 11G,H. All three additional antibodies examined revealed reduced nuclear staining in the second heart field of *Wnt1-Cre; Nedd4^{fl/fl}* embryos, consistent with our initial finding in Figure 4H.

We agree that a Wnt signalling reporter would be an ideal way to demonstrate changes in Wnt signalling activity. However, we are limited by availability of crossing such a reporter to our mouse line, given strict import and quarantine requirements in Australia, which based on our previous experience would require a minimum ~12 months. In lieu of this lengthy timeframe, we now provide additional immunostaining, as well as our prior analysis in Figure 4K that shows downregulation of Wnt-regulated genes specifically in the SHF by in situ hybridisation. Taken together, our data provide multiple lines of evidence to support reduced Wnt signalling in the SHF of *Wnt1-Cre; Nedd4^{fl/fl}* embryos.

6. Could the authors discuss why enhancing Wnt signaling with CHIR99021, a well-known Wnt pathway activator, failed to rescue the OFT defects in *Wnt1-Cre; Nedd4^{fl/fl}* embryos? Could the author analyse Wnt signalling activity under CHIR99021 and iCRT3 conditions?

We hypothesise several possible reasons that CHIR99021 treatment failed to rescue OFT defects in *Wnt1-Cre; Nedd4^{fl/fl}* embryos. One reason may be that the effect of precocious DKK1 overexpression in neural crest cells is too strong to be overcome by increasing canonical Wnt signalling via CHIR99021. Another reason may be that systemic CHIR99021 treatment may cause pleiotropic deleterious effects due to precocious Wnt signalling activation in the whole embryo. Given this, we chose to use a genetic approach (*Dkk1^{+/-}* cross) to try to assess rescue of embryonic phenotypes.

We have now also analysed Wnt signalling activity in iCRT3 treated embryos by assessing β -catenin non-phospho S45 and β -catenin phospho Y489 immunostaining at E9.5 (Supp. Fig. 14). This revealed a reduction in Wnt signalling in the second heart field of iCRT3 treated embryos, and importantly caused a significant reduction in Wnt signalling in *Wnt1-Cre; Nedd4^{fl/+}* (Het) embryos such that these were indistinguishable from *Wnt1-Cre; Nedd4^{fl/fl}* (mutant) embryos. This is consistent with the notion that premature second heart field differentiation can be induced by inhibition of Wnt signalling in genetically susceptible embryos (eg. *Wnt1-Cre; Nedd4^{fl/+}* (Het)), and

strengthens our conclusion that part of the phenotype arises from altered Wnt signalling.

7. In Figure 4G, there appears to be no upregulation of MF20 in *Wnt1-Cre; Nedd4^{fl/fl}* embryos. Could the authors clarify this inconsistency?

We now show an adjacent tissue section for the *Wnt1-Cre; Nedd4^{fl/fl}* embryo from the same experiment, which demonstrates upregulation of MF20, and is a truer representative image of the MF20 staining observed in embryos of this genotype. Upon close examination we found the original image in Fig 4G had a small DAPI-negative area in this region, indicating a small piece of second heart field tissue was torn as a sectioning artefact, explaining the loss of MF20 staining.

8. In Figure 4G, the concept of an "EdU migration front" is unclear. How is this defined, and how does it relate to the extent of SHF incorporation into the myocardium?

We have added a paragraph in the methods to describe how the EdU migration front is defined and measured:

"For EdU migration front distance at E10.5 after 24h chase, the EdU +ve migration front in the OFT was measured as the distance (in μm) of the last EdU +ve cell in the MF20+ve OFT from the pericardial wall. 3-4 sagittal sections were quantified per embryo, and averaged. Individual points on graphs represent the mean value calculated from 1 embryo. A minimum of 5 and up to 6 embryos per genotype were used for quantitation. EdU +ve migration front distance was also normalised to atrio-ventricular valve primordium thickness (an unaffected structure present in the same microscopy images) as a means to account for any variation in embryo size, and is presented in the Supplementary figures."

EdU labelling is able to be utilised as a form of lineage tracing to assess incorporation of SHF cells into the OFT myocardium (eg. see references 17 and 18 cited in text). This is possible owing to the SHF being highly proliferative which readily incorporates EdU, while the OFT is essentially quiescent, and does not take up EdU label. Hence, after a 1hr pulse of EdU which only labels the SHF, any EdU+ve cells in the OFT after a 24h chase are derived from the earlier labelled cells in the SHF. Given that SHF cells move progressively from the SHF into the OFT, the extent of EdU 'migration' distance into the OFT can be used as a measure to assess the extent of SHF cell deployment.

9. Lastly, could the authors discuss the phenotypic differences between *Nedd4K766R* embryos and *Nedd4^{fl/fl}* embryos, and explain why the K766R mutation, which replicates a human variant, does not phenocopy neural crest-specific loss of *Nedd4*?

There are many possible complex reasons why the *Nedd4K766R* mutation in mice does not phenocopy the neural crest-specific loss of *Nedd4*, or additionally the phenotype of *Nedd4^{-/-}* mice. The *Nedd4K766R* variant represents a single amino acid change. Given *Nedd4K766R* mice at E15.5 and E17.5 are grossly phenotypically normal compared to their wildtype littermates (Figure 7E), while *Nedd4^{-/-}* mice are severely growth restricted with notable cranial defects (Cao et. al. 2010, Science Signaling 1:ra5; Wiszniak et. al.

2013, *Developmental Biology* 383:186-200), this indicates Nedd4K766R is not a complete loss-of-function mutation. While Nedd4K766R has lost ability to ubiquitinate DKK1, it is likely that this variant maintains other functional abilities, and has perhaps lost or even altered function in respect to a subset of ubiquitinated targets. Hence, it is likely this variant still maintains some functional capacity in neural crest cells, and hence why this does not phenocopy full loss of Nedd4 function in neural crest cells in *Wnt1-Cre; Nedd4^{fl/fl}* mice.

We now include an additional sentence in our discussion to highlight phenotypic differences between *Nedd4^{K766R}* and *Nedd4^{fl/fl}* mice:

*“While the heart defects exhibited in *Nedd4^{K766R/K766R}* mice do not fully phenocopy the defects observed in *Wnt1-Cre; Nedd4^{fl/fl}* mice, this is likely due to functional differences between hypomorphic *NEDD4 K766R* protein expression in the whole embryo versus complete removal of *NEDD4* protein in neural crest cells of *Wnt1-Cre; Nedd4^{fl/fl}* mice.”*

Reviewer #2 (Remarks to the Author):

In the manuscript “Neural crest cell-derived DKK1 modulates Wnt signalling in the second heart field to orchestrate cardiac outflow tract development, “Wiszniaik et al examine the role of NEDD4 expressed in neural crest cells during cardiac development. The authors demonstrate that the deletion of NEDD4 results in defects in the development of the cardiac outflow tract and the positioning of the great arteries relative to the right and left ventricles. The authors examine the evidence that this effect is due to increased levels of Dkk1 in their mutants. While the idea is very interesting, the data does not support the authors’ conclusions. In many experiments, the experimental setup is not appropriate for what the authors intended to study.

Major Concerns:

1) A major concern is that for multiple types of experiments, normalization is not provided.

We have now added a sentence in the methods under ‘Mice’ to clarify that normalisation was an important consideration for all analyses performed in our manuscript, and that multiple methods were used to ensure stage-matched littermates were examined. *“In all experiments, stage-matched embryo littermates were selected for comparison using stage-appropriate parameters such as somite number, crown-rump length, and forelimb/digit maturity.”* Furthermore, to clarify for the reviewer, we do not observe any developmental delay with *Wnt1-Cre; Nedd4^{fl/fl}* embryos relative to their wildtype littermates, with embryos indistinguishable from each other in terms of size and somite numbers. This can be observed in examples throughout the manuscript such as in Fig. 3A, Fig. 4E, Supp. Fig. 2 and Supp. Fig. 5.

We agree that the reviewers suggestions for additional normalisation for specific experiments is warranted, and now provide these below.

a) For example, the length of the outflow tract is not normalized to an unaffected embryonic structure (e.g., otic vesicle) to control for embryo size in Figure 2A. In addition, it is not clear how authors consistently found a boundary between the proximal OFT and the right ventricle. This needs to be described. Alternatively, the RV needs to be included in the measurements because it is difficult to find such a boundary, and customarily, the RV is included in the measurements of OFT length at E10.5.

As suggested, we have now included the RV in the measurements of OFT length, and present a new quantified figure for OFT+RV length in Figure 2A. Furthermore, we have also normalised the OFT+RV length to an unaffected structure (eye diameter) which we had previously acquired in whole embryo images of the same samples, now presented in Supp. Fig. 2. Both the 'absolute' OFT+RV length and the 'normalised' OFT+RV length demonstrate the same outcome, with reduced OFT+RV length in *Wnt1-Cre; Nedd4^{fl/fl}* embryos, with a p-value of <0.0001. We prefer to present the 'absolute' OFT+RV length in the main figure, as we believe the length in μm is more informative to the reader than the arbitrary ratio of OFT+RV / eye.

b) Another example is the measurement of valve diameter in Figure 5G. This measurement also needs to be normalized to an unaffected embryonic structure to control for embryo size. These normalizations are essential since no single embryo develops at the same rate, and differences in embryo stages and sizes are common even within the same litter.

We have now normalised all of the valve and OFT measurements in Figure 5G to an unaffected structure, that being the neural tube (Supp. Fig. 15D,E). The neural tube was present in all tissue sections for which the valves and OFT were measured. We have now measured the medial neural tube diameter (averaged from 3x sections) for each embryo examined. We present all normalised measurements in Supp. Fig. 15A,B,C, which all demonstrate the same outcomes as the 'absolute' measurements presented in Figure 5 I-K. We prefer to present the 'absolute' values in the main figure, as we believe the diameter in μm or area in μm^2 is more informative to the reader than the arbitrary ratio of diameter / neural tube.

c) It is not proper to report fluorescence intensity as a measurement without normalization, since fluorescence intensity will vary based on many factors beyond the experimenter's control, such as fluctuations in laser power, differences in cover slip thickness, and the exact plane of focus. Check the Bioimaging guide for best practices (https://www.bioimagingguide.org/03_Image_analysis/Intensity.html)

We have now added a section to the methods 'Fluorescence intensity and image quantitation', in which we clarify our methods for Mean Fluorescence Intensity (MFI) measurements, and other quantitation methods. We thank the reviewer for providing the suggested link for best practices in image analysis, of which we address in this comment and in the further comments below.

All images presented throughout the manuscript, and from which quantitation was performed, represent maximum intensity projections of 3x acquired z-stacks per image to account for differences in the exact focal plane. Images were always acquired in the same microscopy session on the same day with the same settings for paired analyses to minimise any variances in laser power or PMT sensitivity/acquisition. Images were carefully acquired such that no saturated pixels were present, and the raw (.czi) files were always used for quantitative analysis. The fluorescence intensity measurements presented are the 'mean fluorescence intensity' measurements; ie. the absolute fluorescence in the region of interest is normalised to the area of the region of interest. Equivalent areas of 'region of interest' were selected (as now described in greater detail in our methods, as well as indicated in our figures, eg. Supp. Fig. 6).

Importantly, we have carefully selected our representative images in all figures to demonstrate what we observe many times over for individual analyses. We use MFI as a way to quantify this for the purpose of statistical analysis to confirm and add power to our visual observations.

Where possible, we have now incorporated additional normalisations into our MFI measurements, as documented below.

2) Lines through the ventricles seem to be drawn in random orientation in the mutants in Figure 5. While in controls, the line is drawn through the lumens of LV and RV, in the mutants, the line does not appear to go through the lumen of LV. Other section levels or coronal sections may be more appropriate. Multiple levels like those shown in Fig. 2B may also work well to demonstrate altered alignment.

We now show multiple levels of sections (like for Fig. 2B) to better demonstrate differences in OFT rotation, and present these in Supp. Fig. 13. All additional sections shown are from the original embryos shown in Fig. 5B and 5H, except for the vehicle treated *Wnt1-Cre; Nedd4^{fl/fl}* embryo in 5B as the section through the valve was damaged. We have picked another littermate imaged in the same experiment that had complete and undamaged sections for the artery/valve/ventricle, and now present these in both Fig. 5B and Supp. Fig. 13A.

3) On line 14, page 6, the authors state that NC numbers are not affected in the mutants. Since this is very important for the rest of the study, the authors need to provide quantification to support this statement.

We now provide data to support this statement and present neural crest cell number quantitation in Supp. Fig. 3, with no significant differences between genotypes.

4) On line 19, page 6, the authors state that the expression of additional SHF markers were not affected in the mutants. However, Sup. Fig 2 shows that *Hoxb1* and *BMP4* are downregulated and *Fgf8* is upregulated in the mutants. Please state how many times each experiment was conducted; how many controls and mutants were analyzed for each staining, and provide quantification.

We have now included higher magnification images of the second heart field region of the whole-mount embryos, to more clearly demonstrate the region of interest that was

assessed when comparing wildtype versus *Wnt1-Cre; Nedd4^{fl/fl}* embryos, and show this in Supplementary Figure 5. Furthermore, we have performed semi-quantitative analysis of BCIP/NBT substrate staining in the SHF region from the multiple experiments that were performed. At least 3 (and up to 7) embryos were analysed per probe per genotype. We did not observe any significant differences in gene expression in the SHF region between wildtype and *Wnt1-Cre; Nedd4^{fl/fl}* embryos. We have now added the following paragraph to our methods section:

“Semi-quantitative optical density measurements of whole-mount in situ hybridisation were performed in ImageJ. Images were subjected to colour deconvolution to generate 8-bit grayscale images of BCIP/NBT substrate staining. Mean grey values were calculated from a region of interest corresponding to the second heart field/dorsal pericardial wall. Optical density was calculated using the formula $OD = \text{Log}_{10}(255/\text{mean grey value})$. A minimum of 3 and up to 7 whole mount embryos per genotype were assessed.”

5) For the experiment with the explant in Fig. 3E, FACS is more appropriate to determine the proportion of MF20+ cells and intensity. Western blot is also appropriate, but the measurements shown are not up to the accepted standards for quantification.

We agree that flow cytometry would provide more quantitative measures of MF20+ve cell proportions and staining intensity. However, based on other current work in our laboratory (work up of E9.5 SHF cell dissociation for scRNAseq, yielding around 2000 cells per SHF tissue) it is not possible to dissociate a single SHF in individual tubes, and have sufficient cells for performing subsequent analyses (in this case, fixing, permeabilising and staining cells for MF20, then flow cytometry analysis). We politely disagree that western blotting for MF20 is more appropriate than immunostaining. Western blotting would require scraping the SHF tissue explant from the growth substrate, potentially leaving behind or damaging cells such there will be inherent differences between samples. Sample integrity and the consistency between samples is much better maintained by simply fixing the tissue in place with PFA. Staining for the MF20 antigen via western blot with primary and secondary antibody is equivalent to staining the MF20 antigen via immunofluorescence with primary and secondary antibody. Then the principles for quantitation of both methods are similar (densitometry of a grayscale image). We have also now included further information on our method of quantitation: *“For MF20 MFI measurements of SHF explants, the entire DAPI+ve region was selected, and MFI measured in this region of interest. Individual values represent MFI for independent explants (ie. biological replicates) of the indicated genotype.”*

Given we already show MF20 is upregulated *in vivo*, the SHF explant experiments are intended to be confirmatory, and do not change the conclusions of our paper. If the reviewer does not agree our quantitation methods for this experiment are valid, we are satisfied to remove the graph in Figure 3E and simply show the representative immunofluorescence image of each genotype.

6) The EdU experiment in Fig. 3G is not appropriate to show cell migration. For this experiment, EdU pulse-chase needs to be implemented: after one hour of EdU,

thymidine needs to be injected to compete EdU off. Otherwise, the experiment in 5G can be interpreted as a decrease in proliferation rather than migration. See doi: 10.1016/j.ydbio.2016.02.017 as an example of such an assay in vivo. Also see papers by Elaine Fuchs, as additional examples for how this is done.

We have now performed this EdU pulse-chase experiment as per the reviewers suggestion, and present results in an updated Figure 3G and Supp. Fig. 7. We have also updated our methods to reflect the 1hr EdU pulse, followed by thymidine injection, and then a 24h chase.

Importantly, the pulse-chase approach confirmed our original findings, with reduced migration of EdU labelled cells into the outflow tract in *Wnt1-Cre; Nedd4^{fl/fl}* embryos. We also normalised the EdU migration distance to an unaffected structure (A-V valve thickness) to account for any variability in embryo size.

7) Figure 4H. There is no nuclear β -catenin in controls or mutants. It is not clear what is being measured and how.

We now present higher magnification images of β -catenin immunostaining in Supplementary Figure 11. Furthermore, we have now highlighted examples of nuclear localised β -catenin (solid arrowheads) vs examples of no nuclear β -catenin (outlined arrowheads) to make our quantitation methods clearer for the reader.

8) Figure 4K: Expression of Axin2 can be used as a readout for Wnt signaling. However, the authors point to Axin2 staining in the atrial chamber in a control embryo (arrowhead) and not the SHF in the dorsal pericardial wall contiguous with the OFT. The SHF appears not to express Axin2 in the control. In the mutant shown, the Axin2 staining did not seem to work because there is no staining in the atrial chamber, a tissue unaffected by the neural crest. This is an important experiment, and it needs to be done properly to determine whether Wnt signaling is indeed downregulated in the SHF in the mutants.

We now provide uncropped images of the *Axin2* in situ hybridisation presented in Fig. 4K in Reviewer Figure 3. This demonstrates that the *in situ* procedure worked well, as *Axin2* expression is evident in unaffected structures such as pharyngeal arch 1 and the posterior pharyngeal endoderm. We also use dashed lines to indicate the boundaries of the dorsal pericardial wall, outflow tract, and atrial chamber, which highlight expression of *Axin2* in the dorsal pericardial wall in the *wildtype* section, but not in the *Wnt1-Cre; Nedd4^{fl/fl}* section.

9) Figure 5E: MF20 IF intensity is reported without normalization and thus is unreliable. In addition to properly measuring and normalizing MF20 intensity, the authors should measure the length of SHF occupied by the ectopic MF20+ cells.

While our prior analysis was carefully performed with bioimaging best practices considered (as described in response to point 1c above), we have now additionally performed further normalisation as per the reviewers suggestion. The atrial chamber, which is also positive for MF20 immunostaining, is considered an unaffected structure

in our analyses, given neural crest cells do not make contact with or influence its development at the stage assessed (E9.5). The atrial chamber is also adjacent to the second heart field in sagittal sections, and as such was acquired in all images used previously to quantify MF20 mean fluorescence intensity in the second heart field. We have now measured fluorescence intensity in a region of the atrial wall (as described in our methods and shown in Supp. Fig. 6A), and have used this to normalise our previously measured MFI measurements in the SHF. We have performed normalisation for all MF20 MFI measurements in the SHF shown in Fig. 3D and 5E, and present these in Supp. Fig. 6C and Supp. Fig. 6E. Furthermore, we have also measured the length of SHF occupied by the ectopic MF20-positive cells, and present this in Supp. Fig. 6D and Supp. Fig. 6F. Importantly, all analysis methods revealed statistically significant differences in MF20 immunostaining in the SHF.

10) Figure 6: How do authors explain the mechanism by which NEDD4 regulates the ubiquitination of Dkk1? Dkk1 is a secreted protein (i.e., it is present inside the ER, Golgi, and secretory vesicles), while NEDD4 is a cytoplasmic protein that is always outside the ER, Golgi, etc. These two proteins are never in the same cellular compartment.

We now show 63x high magnification images of cardiac neural crest cells *in vivo* immunostained for NEDD4, DKK1 and GM130 (a cis-Golgi marker) in Figure 6F. This demonstrates co-localisation of NEDD4 and DKK1, suggesting it is feasible that NEDD4 could make contact with DKK1 in the same cellular compartment.

Prior publications also provide evidence for roles for NEDD4 in other cellular compartments such as Golgi and endosomes, including some from the current authorship team of this manuscript.

Eg:

<https://www.nature.com/articles/s12276-025-01396-2> (Shows NEDD4 ubiquitinates key golgi protein GM130).

<https://www.nature.com/articles/s41467-025-57944-x> (Shows Nedd4(NE) in endosomes. Our own unpublished work has also discovered this NE version of Nedd4 is present in mouse too, not just 'primate' specific as per the paper title.)

<https://www.sciencedirect.com/science/article/pii/S0006291X21005623?via%3Dihub> (Shows Nedd4 interaction helps traffic a protein from Golgi to late endosomes).

<https://www.sciencedirect.com/science/article/pii/S0021925819363045?via%3Dihub> (Paper from the authorship research team, showing ubiquitination of a Golgi protein (Ndfip2) by Nedd4).

11) Figure 7G. The hearts look smaller in the mutants, suggesting that VSD and other “defects” may be due to delayed embryo development at E15.5 in the mutants. E18.5 is a more appropriate time point for the evaluation of structural heart defects.

a. Are Nedd4-K766R homozygous mutants viable?

Yes, indeed the hearts are smaller in *Nedd4*^{K766R/K766R} mutants. This is observed at both E15.5 and E17.5 (Figure 7G,H), and we have now added example images for E17.5 to the main figures of our manuscript to show these defects persist at older stages of

development. Our data is consistent with this being a specific defect and not likely due to developmental delay, given *Nedd4*^{K766R/K766R} embryos are equivalent in size to their wildtype littermates and gross embryo development appears normal (Figure 7E). We evaluated structural heart defects over a range of stages at late gestation from E15.5 – E17.5, with the defects listed in Figure 7F present across all ages examined. We chose E15.5 onwards as several other studies have reported this to be an appropriate time to phenotype cardiac defects as all major structures of the heart have formed by this stage (eg. Schneider et. al. 2004 BMC Dev Biol 4:16; Liu et. al 2013 Circulation: Cardio. Im. 7:31; Kim et. al. 2013 Circulation: Cardio. Im. 6:551.) E15.5 onwards is also considered appropriate to assess VSDs without concern for developmental delay, as the interventricular septum has normally completed development by E13.5 (eg. Anderson et. al. 2014 Anat. Rec. 297:1414; Webb et. al. 1998 Circulation 82:645; Krishnan et. al. 2014 Pediatric Res. 76:500). Limitations in terms of compliance with University of South Australia Animal Ethics Committee guidelines have precluded pups being born from *Nedd4*^{K766R/+} x *Nedd4*^{K766R/+} crosses, hence we have not yet been able to assess if *Nedd4*^{K766R/K766R} mutants are viable post-birth.

Minor:

1) on line 14 page 4, the authors state that “neural crest cells form structural components of the outflow tract valves.” This is incorrect. Neural crest cells die at late gestation and do not contribute to mature outflow tract valves. See papers from Jonathan Epstein lab.

We have now revised this sentence to reflect that neural crest cells form components of the heart during the early development stages, even though they undergo apoptosis in some structures (such as the valves) later in developmental maturation.

“In the developing heart, neural crest cells form ~~structural~~ components of the outflow tract valves, arteries, and conotruncal septum.”

2) On line 19-21 page 4, the authors state “However, prior models in which neural crest cells have been surgically or genetically ablated⁴⁻⁷ have not been amenable to elucidate how neural crest cells interact with the second heart field to control its growth, differentiation and morphogenesis.” This is inaccurate. The authors should mention the work from Margaret Kirby’s lab showing the importance of NC in regulating BMP4-FGF8 cross talk between NCCs and SHF in preventing precocious differentiation of cardiac progenitors in the SHF.

Our intention with this sentence was to emphasise that these prior models have completely ablated the neural crest cells, hence it is not possible to assess how these neural crest cells would have normally signalled to or interacted with the second heart field (as they are now not even present to assess). We wholeheartedly agree that these ablation models were critical to discover that neural crest cells play important roles in heart development, and do indeed cite how these models have uncovered roles for BMP-FGF signalling and cross-talk in our discussion section.

We have modified our statement: *“However, prior models in which neural crest cells have been surgically or genetically ablated⁴⁻⁷ have not been amenable to elucidate how*

neural crest cells normally interact with the second heart field to control its growth, differentiation and morphogenesis.”

Reviewer #3 (Remarks to the Author):

The MS by Wiszniak et al reports a novel regulatory mechanism whereby the ubiquitin ligase NEDD4 affects the protein levels of DKK1 in neural crest cells that modulates Wnt signaling thereby affecting the timing of second heart field differentiation and outflow tract morphogenesis. A combination of in vivo genetic studies in mice and in vitro examination of NEDD4 function provide convincing evidence for the major conclusions of the study. Notably, a DKK1 deficient mouse provides evidence for a rescue of the NEDD4 deficiency in neural crest cells which is important support for the proposed mechanism. In addition, variants in NEDD4 were identified in individuals with congenital heart malformations and loss of function demonstrated experimentally. This is a strong study with convincing data to support a new regulatory contribution of Wnt signaling in cardiac outflow tract development with human clinical implications.

Minor comments

1. The title does not mention NEDD4 which is a major focus of the study.

We have now included NEDD4 in the title.

2. Were abnormalities in outflow tract endocardial cushions or semilunar valve primordia observed?

Yes, we do indeed observe abnormalities in the morphology of the endocardial cushions of the semilunar valves, as can be seen in Figure 2B. While we alluded to this in the Figure Legend “*Valve leaflet defects are also observed*”, we did not include this in the main manuscript text. We have now briefly mentioned this in the main text:

“Semilunar valve morphology was also altered (Fig. 2B).”

We believe this is a very interesting phenotype that is independent of the second heart field differentiation mechanism, and is currently being investigated in a separate project in our laboratory, and will be the focus of a new manuscript in the near future.

3. The Discussion is long and includes information on BMP and FGF signaling that seem peripheral to the current study.

We agree that the discussion regarding additional signalling pathways is somewhat peripheral to the current study that is focused on Wnt signalling. However, we have decided to maintain the information regarding BMP and FGF signalling as this was a specific request for inclusion from Reviewer 2.

Reviewer Figure 1

A Human iPSC-derived NCCs day10 differentiation

B

Reviewer Figure 1 - Building an *in vitro* model of cardiac neural crest cells

A: Human iPSCs were directed to differentiate into neural crest cells in our established unpublished protocol (briefly, SMAD inhibition + Wnt induction, followed by FGF2, BMP4, RA treatment for 10 days). scRNAseq reveals broad induction of NCC markers such as *TFAP2A* and *SOX10*. Cluster 12 is *SOX10*-ve and *DKK1*+ve, consistent with expression profile of cardiac neural crest cells at E9.5. This population is also +ve for other cardiac NCC markers such as *HAND2*, *HAND1* and *GATA3*. **B:** Immunostaining reveals a small proportion of these cells express *DKK1* (arrowheads). **C:** scRNAseq of E9.5 *Wnt1-Cre;EGFP* FACS-sorted neural crest cells. *Dkk1* is specifically expressed in cluster 14, along with other cardiac neural crest cell markers. Importantly, this highlights *Dkk1* as a novel cardiac neural crest cell marker.

C Mouse E9.5 *Wnt1-Cre;EGFP* +ve NCCs

From: De Bono et. al (2023) Single-cell transcriptomics uncovers a non-autonomous *Tbx1*-dependent genetic program controlling cardiac neural crest cell development. *Nature Communications* 14:1551

Reviewer Figure 2

Reviewer Figure 2 - *Nedd4*^{-/-} embryos have severe growth restriction and vascular defects

Example images of wildtype and *Nedd4*^{-/-} embryos at E11.5 and E12.5. *Nedd4*^{-/-} embryos exhibit growth restriction from around E11.5 onwards, and have severe cardiovascular defects.

Reviewer Figure 3

Reviewer Figure 3 - *Axin2* *in situ* hybridisation

Uncropped images of *in situ* hybridisation for *Axin2* as shown in Figure 4K. Expression is evident in other areas such as the pharyngeal arch and posterior pharyngeal endoderm.

Dashed lines in lower panels show overlay of tissue boundaries.

pa1, pharyngeal arch 1; pe, pharyngeal endoderm; OFT, outflow tract; dpcw, dorsal pericardial wall; A, atrium; V, ventricle.

Response to reviewers' comments:

REVIEWERS' COMMENTS

Reviewer #1 (Remarks to the Author):

I am satisfied with the authors' response to my previous comment. Overall, this is an insightful and compelling study with strong genetic evidence. For clarity, I would still recommend moving Supplementary Figure 4A and 4B into the main figures.

We have now moved both Supplementary Figure 3 and 4 into the main figures, as new panels in Figure 3 (now Fig. 3C-E).

Reviewer #2 (Remarks to the Author):

The revisions of the manuscript entitled "Neural crest cell derived DKK1 and NEDD4 modulate Wnt signalling in the second heart field to orchestrate cardiac outflow tract development," has resulted in a really beautiful manuscript with high-quality data reporting novel and interesting findings that are also relevant to human congenital heart disease. The revised manuscript has addressed all previous critiques. The following are minor concerns remaining:

1) Reviewer Figure 3 - Axin2 in situ hybridization should be included in the manuscript.

We now include this as Supp. Fig. 11.

2) Please mention which Wnt1-Cre strain was used? Wnt1-Cre1 or Wnt1-Cre2? Wnt1-Cre1 ectopically expresses large amounts of Wnt1 protein (Lewis AE, Vasudevan HN, O'Neill AK, Soriano P, & Bush JO (2013). The widely used Wnt1-Cre transgene causes developmental phenotypes by ectopic activation of Wnt signaling. *Dev Biol*, 379(2), 229–234. 10.1016/j.ydbio.2013.04.026).

The Wnt1-Cre strain used (Wnt1-Cre1) is listed in our methods (ref. 51). This is the gold-standard line for the field. As per the reference included above, and from our own and other researchers' observations in the neural crest field, the Wnt1-Cre1 line only causes significant adverse effects when maintained in a Wnt1-Cre^{tg/tg} homozygous state. We only ever use this line in a hemizygous Wnt1-Cre^{tg/+} state.

3) In the methods, please add the genetic background of each parental strain used. In general, it would be useful to know the parental genotypes of the crosses that were used to generate various embryo genotypes. This information should be added to the methods as well.

Please add to the methods whether Wnt1-Cre; Nedd4fl/fl; Dkk1^{+/-} embryos used for quantifications in Fig. 5G are littermates of Wnt1-Cre; Nedd4fl/fl; Dkk1^{+/+}. If they were not littermates, please note whether or not differences in phenotypes between Wnt1-Cre; Nedd4fl/fl; Dkk1^{+/-} and Wnt1-Cre; Nedd4fl/fl; Dkk1^{+/+} may be due to potential differences in genetic background.

All embryos used for analyses in Fig. 5G-L were littermates. Our experiment was carefully designed this way to eliminate any potential effects due to genetic background. Embryos were derived from litters of *Wnt1-Cre; Nedd4^{fl/+}* males crossed to *Nedd4fl/fl; Dkk1^{+/-}* females, yielding embryos of all listed genotypes.

We now add this to the methods section:

“*Wnt1-Cre; Nedd4^{fl/fl}* mice were intercrossed with *Dkk1^{+/-}* mice (*Wnt1-Cre; Nedd4^{fl/+}* males crossed to *Nedd4^{fl/fl}; Dkk1^{+/-}* females) to generate *Wnt1-Cre; Nedd4^{fl/fl}; Dkk1^{+/-}* embryos (as well as littermate controls) for assessing the rescue effect of DKK1 dosage compensation.”

4) Please add a reference to your EdU pulse-chase method.

We now add references to the methods section:

“In methods similar to ^{17,18}, pregnant dams were intraperitoneally injected with 100 mg/kg EdU (5-ethynyl-2'-deoxyuridine) at E9.5 to pulse-label proliferating cells.”

5) Sup. Fig. 8: pErk1/2 in staining in Sup. Fig. 8 is too dim and is hard to see.

We have increased the brightness of all panels in pERK1/2 figure (now Supp. Fig. 6B).